# Tracking prototype and exemplar representations in the brain across learning

Caitlin R Bowman[1,2]*, Takako Iwashita[1], Dagmar Zeithamova[1]*

[1]Department of Psychology, University of Oregon, Eugene, United States; [2]Department of Psychology, University of Wisconsin-Milwaukee, Milwaukee, United States

**Abstract** There is a long-standing debate about whether categories are represented by individual category members (exemplars) or by the central tendency abstracted from individual members (prototypes). Neuroimaging studies have shown neural evidence for either exemplar representations or prototype representations, but not both. Presently, we asked whether it is possible for multiple types of category representations to exist within a single task. We designed a categorization task to promote both exemplar and prototype representations and tracked their formation across learning. We found only prototype correlates during the final test. However, interim tests interspersed throughout learning showed prototype and exemplar representations across distinct brain regions that aligned with previous studies: prototypes in ventromedial prefrontal cortex and anterior hippocampus and exemplars in inferior frontal gyrus and lateral parietal cortex. These findings indicate that, under the right circumstances, individuals may form representations at multiple levels of specificity, potentially facilitating a broad range of future decisions.

*For correspondence:
cbowman@uoregon.edu (CRB);
dasa@uoregon.edu (DZ)

**Competing interests:** The authors declare that no competing interests exist.

## Introduction

The ability to form new conceptual knowledge is a key aspect of healthy memory function. There has been a longstanding debate about the nature of the representations underlying conceptual knowledge, which is exemplified in the domain of categorization. Some propose that categories are represented by their individual category members and that generalizing the category label to new examples involves joint retrieval and consideration of individual examples encountered in the past (i. e., exemplar models, *Figure 1A*; *Kruschke, 1992*; *Medin and Schaffer, 1978*; *Nosofsky, 1986*). Others propose that categories are represented by their central tendency – an abstract prototype containing all the most typical features of the category (i.e., prototype models, *Figure 1B*; *Homa, 1973*; *Posner and Keele, 1968*; *Reed, 1972*). Category generalization then involves consideration of a new item's similarity to relevant category prototypes.

Both the prototype and exemplar accounts have been formalized as quantitative models and fit to behavioral data for decades, with numerous studies supporting each model (exemplar meta-analysis: *Nosofsky, 1988*; prototype meta-analysis: *Smith and Minda, 2000*). Neuroimaging studies have also provided support for these models. Studies using univariate contrasts showed overlap between neural systems supporting categorization and recognition (*Nosofsky et al., 2012*), as well as medial temporal lobe involvement in categorization (*Koenig et al., 2008*; *Lech et al., 2016*; *Nomura et al., 2007*), both of which have been interpreted as indicating a role of exemplar retrieval in categorization. More recently, studies have used parameters generated from formal prototype and exemplar models with neuroimaging data, but with conflicting results. *Mack et al., 2013* found similar behavioral fits for the two models, but better fit of the exemplar model to brain data. Parts

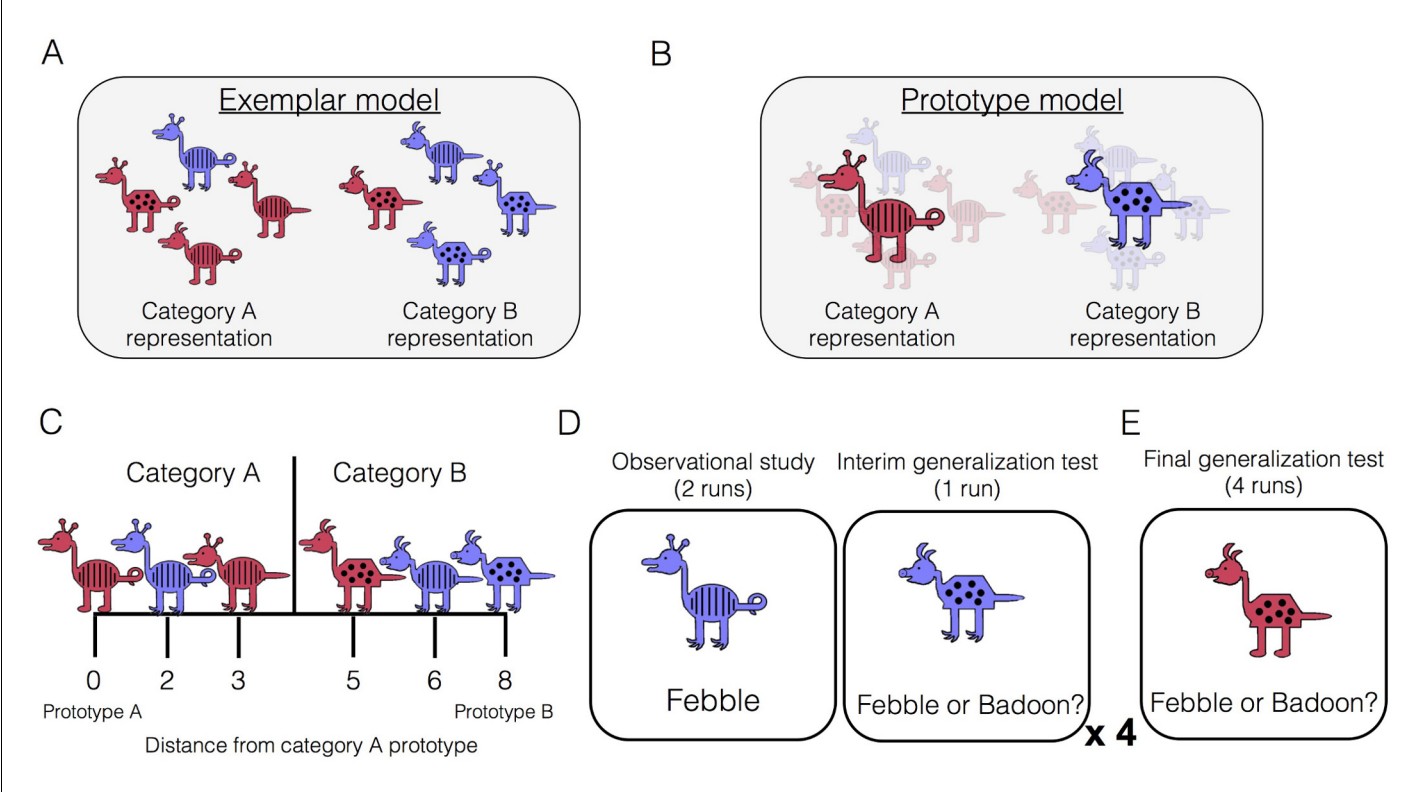

**Figure 1.** Category-learning task. Conceptual depiction of (A) exemplar and (B) prototype models. Exemplar: categories are represented as individual exemplars. New items are classified into the category with the most similar exemplars. Prototype: categories are represented by their central tendencies (prototypes). New items are classified into the category with the most similar prototype. (C) Example stimuli. The leftmost stimulus is the prototype of category A and the rightmost stimulus is the prototype of category B, which shares no features with prototype A. Members of category A share more features with prototype A than prototype B, and vice versa. (D) During the learning phase, participants completed four study-test cycles while undergoing fMRI. In each cycle, there were two runs of observational study followed by one run of an interim generalization test. During observational study runs, participants saw training examples with their species labels without making any responses. During interim test runs, participants classified training items as well as new items at varying distances. (E) After all study-test cycles were complete, participants completed a final generalization test that was divided across four runs. Participants classified training items as well as new items at varying distances.

of the lateral occipital, lateral prefrontal and lateral parietal cortices tracked exemplar model predictors. No region tracked prototype predictors. The authors concluded that categorization decisions are based on memory for individual items rather than abstract prototypes. In contrast, *Bowman and Zeithamova, 2018* found better fit of the prototype model in both brain and behavior. The ventromedial prefrontal cortex and anterior hippocampus tracked prototype predictors, demonstrating that neural category representations can involve more than representing the individual category members, even in regions like the hippocampus typically thought to support memory for specific episodes.

Interestingly, the different brain regions identified across these two studies aligned well with the larger literature contrasting memory specificity with memory integration and generalization. Lateral prefrontal regions are thought to resolve interference between similar items in memory (*Badre and Wagner, 2005*; *Bowman and Dennis, 2016*; *Jonides et al., 1998*; *Kuhl et al., 2007*), and lateral parietal cortex supports recollective experience (*Vilberg and Rugg, 2008*) and maintains high fidelity representations of individual items during memory retrieval (*Kuhl and Chun, 2014*; *Xiao et al., 2017*). That these regions also tracked exemplar predictors suggests that these functions may also support categorization by maintaining representations of individual category members as distinct from one another and from non-category members. In contrast, the VMPFC and hippocampus are known to support episodic inference through memory integration of related episodes (*Schlichting et al., 2015*; *Shohamy and Wagner, 2008*; *Zeithamova et al., 2012*) and encoding of new information in light of prior knowledge (*van Kesteren et al., 2012*). That these regions also

tracked prototype predictions suggests that prototype extraction may involve integrating across category exemplars, linking across items sharing a category label to form an integrated, abstract category representation. However, as neural prototype and exemplar representations were identified across studies that differed in both task details and in the categorization strategies elicited, it has not been possible to say whether differences in the brain regions supporting categorization were due to differential strength of prototype versus exemplar representations or some other aspect of the tasks.

It is possible that the seemingly conflicting findings regarding the nature of category representations arose because individuals are capable of forming either type of representation. Prior studies have compared different category structures and task instructions to identify multiple memory systems supporting categorization (e.g., *Aizenstein et al., 2000*; *Ashby et al., 1998*; *Ell et al., 2010*; *Zeithamova et al., 2008*). While such findings show that the nature of concept representations depend on task demands, it is unclear if both prototype and exemplar representations can co-exist within the same task. Such mixed representations have been identified in episodic memory tasks, with individuals sometimes forming both integrated and separated representations for the same events (*Schlichting et al., 2015*) and a single episode sometimes represented at multiple levels of specificity, even within the hippocampus (*Collin et al., 2015*). We also know that individuals sometimes use a mix of strategies in categorization, for example when most category members are classified according to a simple rule while others are memorized as exceptions to that rule (*Davis et al., 2012*; *Nosofsky et al., 1994*). These differing representations may emerge because they allow for flexibility in future decision-making, as abstract representations that discard details of individual items are well suited to making generalization judgments but are poorly suited to judgments that require specificity. Alternatively, prototype representations may emerge as a byproduct of retrieving category exemplars, and they may themselves be encoded via recurrent connections, becoming an increasingly robust part of the concept representation (*Hintzman, 1986*; *Koster et al., 2018*; *Zeithamova and Bowman, 2020*). Thus, under some circumstances, both prototype and exemplar representations may be apparent within the same task.

To test this idea, we used fMRI in conjunction with a categorization task designed to balance encoding of individual examples vs. abstract information. This task used a training set with examples relatively close to the prototype, which has been shown to promote prototype abstraction (*Bowman and Zeithamova, 2018*; *Bowman and Zeithamova, 2020*). To promote exemplar encoding, we used an observational training task rather than feedback-based training (*Cincotta and Seger, 2007*; *Heindel et al., 2013*; *Poldrack et al., 2001*). We then looked for evidence of prototype and exemplar representations in the brain and in behavioral responses. In behavior, the prototype model assumes that categories are represented by their prototypes and predicts that subjects should be best at categorizing the prototypes themselves, with decreasing accuracy for items with fewer shared features with prototypes. The prototype model does not make differential predictions for new and old (training) items at the same distance from the prototype. The exemplar model assumes that categories are represented by the previously encountered exemplars and predicts that subjects should be best at categorizing old items and new items closest to the old exemplars. The mathematical formalizations of the models further take into account that a participant may not pay equal attention to all stimulus features and that perceived distance increases non-linearly with physical distance (see Methods for more details). We note that it is sometimes possible to observe behavioral evidence for both types of representations. For example, in our prior study (*Bowman and Zeithamova, 2018*), participants' behavior was better explained by the prototype model than the exemplar model, but we also observed an advantage for old items relative to new items at the same distance to prototypes, in line with exemplar but not prototype model predictions.

The key behavioral prediction of each model is the trial-by-trial probability of responding category A vs category B. These probabilities are determined for each trial by the relative similarity of the test item to the category A and category B representations proposed by each model. Once these probabilities are generated for each model, they are compared to the participant's actual responses to determine which model better predicted the subject's observed behavior. We also used output from the models to generate subject-specific, trial-by-trial fMRI predictions. These were derived from the similarity of each test item to either an exemplar-based or prototype-based category representation (see Methods for details). We then measured the extent to which prototype- and exemplar-tracking brain regions could be identified, focusing on the VMPFC and anterior

hippocampus as predicted prototype-tracking regions, and lateral occipital, prefrontal, and parietal regions as predicted exemplar-tracking regions.

We also asked whether there are shifts across learning in the type of concept representation individuals rely on to make categorization judgments. While some have suggested that memory systems compete with one another during learning (*Poldrack and Packard, 2003*; *Seger, 2005*), prior studies fitting exemplar and prototype models to fMRI data have done so only during a categorization test that followed extensive training, potentially missing dynamics occurring earlier in concept formation. Notably, memory consolidation research suggests that memories become abstract over time, often at the expense of memory for specific details (*McClelland et al., 1995*; *Moscovitch et al., 2016*; *Payne et al., 2009*; *Posner and Keele, 1970*), suggesting that early concept representations may be exemplar-based. In contrast, research on schema-based memory shows that abstract knowledge facilitates learning of individual items by providing an organizational structure into which new information can be incorporated (*Bransford and Johnson, 1972*; *Tse et al., 2007*; *van Kesteren et al., 2012*). Thus, early learning may instead emphasize formation of prototype representations, with exemplars emerging later. Finally, abstract and specific representations need not trade-off in either direction. Instead, the brain may form these representations in parallel (*Collin et al., 2015*; *Schlichting et al., 2015*) without trade-off between concept knowledge and memory for individual items (*Schapiro et al., 2017*), generating the prediction that both prototype and exemplar representations may grow in strength over the course of learning.

In the present study, participants underwent fMRI scanning while learning two novel categories or 'species,' which were represented by cartoon animals varying on eight binary dimensions (*Figure 1C*). The learning phase consisted of two types of runs: observational study runs and interim generalization test runs (*Figure 1D*). During study runs, participants passively viewed individual category members with their accompanying species label ('Febble' or 'Badoon'). All of the items presented during study runs differed by two features from their respective prototypes (for example, exemplars depicted in *Figure 1A*). After completing two runs of observational study, participants underwent an interim generalization test run in which participants classified cartoon animals into the two species. Test items included the training items as well as new items at varying distances from category prototypes. Across the entire learning phase, there were four study-test cycles, with different new test items at every cycle. The learning phase was followed by a final generalization test, whose structure was similar to the interim test runs but more extensive (*Figure 1E*).

To test for evidence of prototype and exemplar representations in behavior across the group, we compared accuracy for items varying in distance from category prototypes and for an accuracy advantage for training items relative to new items matched for distance from category prototypes. We also fit formal prototype and exemplar models to behavior in individual subjects, which involves computing the similarity of a given test item to either the prototype of each category (prototype model) or the individual training items from each category (exemplar model), which is then used to make predictions about how likely it is that an item will be classified into one category versus the other. The model whose predictions better match a given subject's actual classification responses will have better fit. However, it is also possible that evidence for each of the models will be similar, potentially reflecting a mix of representations.

To test for co-existing prototype and exemplar correlates in the brain during interim and final generalization tests, we used latent metrics generated from each model as trial-by-trial predictors of BOLD activation in six regions of interest (*Figure 2*): ventromedial prefrontal cortex, anterior hippocampus, posterior hippocampus, lateral occipital cortex, inferior frontal gyrus, and lateral parietal cortex. To identify potential changes with learning, we tested these effects separately in the first half of the learning phase (interim tests 1 and 2) and second half of the learning phase (interim tests 3 and 4) as well as in the final test.

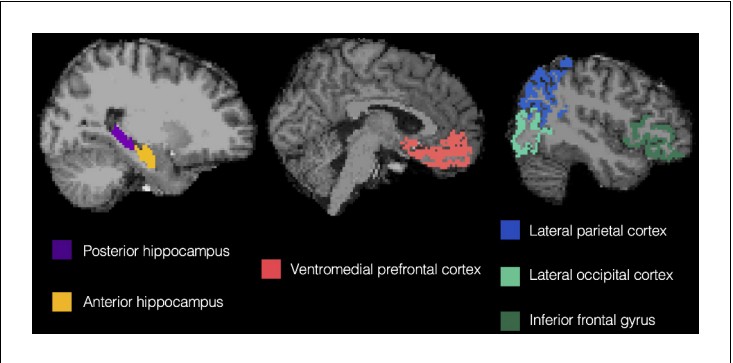

**Figure 2.** Regions of interest from a representative subject. Regions were defined in the native space of each subject using automated segmentation in Freesurfer.

# Results

## Behavioral

### Accuracy

#### Interim tests

Categorization performance across the four interim tests is presented in *Figure 3A*. We first tested whether generalization accuracy improved across the learning phase and whether generalization of category labels to new items differed across items of varying distance to category prototypes. There was a significant main effect of interim test number [$F(3,84)$=3.27, p=0.03, $\eta_p^2$ = 0.11], with a significant linear effect [$F(1,28)$=9.91, p=0.004, $\eta_p^2$ = 0.26] driven by increasing generalization accuracy across the interim tests. There was also a significant main effect of item distance [$F(3,84)$=51.75, p<0.001, $\eta_p^2$ = 0.65] with a significant linear effect [$F(1,28)$=126.04, p<0.001, $\eta_p^2$ = 0.82] driven by better accuracy for items closer to category prototypes. The interim test number x item distance interaction effect was not significant [$F(9,252)$=0.62, p=0.78, $\eta_p^2$ = 0.02]. We next tested whether accuracy for old training items was higher than new items of the same distance (i.e., distance 2) and whether that differed over the course of the learning phase. There was a linear effect of interim test number [$F(1,28)$=16.78, p<0.001, $\eta_p^2$ = 0.38] driven by increasing accuracy across the tests. There was also a significant main effect of item type (old vs. new) [$F(1,28)$=8.76, p=0.01, $\eta_p^2$ = 0.24], driven by higher accuracy for old items (M = 0.83, SD = 0.11) relative to new items of the same distance from the prototypes (M = 0.77, SD = 0.10). The interim test number x item type interaction effect was not significant [$F(3,84)$=0.35, p=0.79, $\eta_p^2$ = 0.01], indicating that the advantage for old compared to new items was relatively stable across learning. To summarize, we observed a reliable typicality gradient where accuracy decreased with the distance from the prototypes and both old and new items at the distance two numerically fell between distance 1 and distance three items (*Figure 3A*). However, within distance two items, we also observed a reliable advantage for the old items compared to new items, an aspect of the data that would not be predicted by the prototype model.

#### Final test

Accuracies for generalization items at each distance from the prototype as well as for training items (all training items were at distance two from the prototypes) are presented in *Figure 3B*. A repeated measures ANOVA on new items that tested the effect of distance from category prototypes on generalization accuracy showed a main effect of item distance [$F(3,84)$=53.61, p<0.001, $\eta_p^2$ = 0.66] that was well characterized by a linear effect [$F(1,28)$=124.55, p<0.001, $\eta_p^2$ = 0.82]. Thus, the categorization gradient driven by higher accuracy for items closer to category prototypes observed during learning was also strong during the final test. In contrast, a paired t-test for accuracy on old relative to new items at distance two showed that the numeric advantage for old relative to new items was not statistically significant in the final test [$t(28)$=0.93, p=0.36, $CI_{95}$[−0.03,.08], d = 0.22].

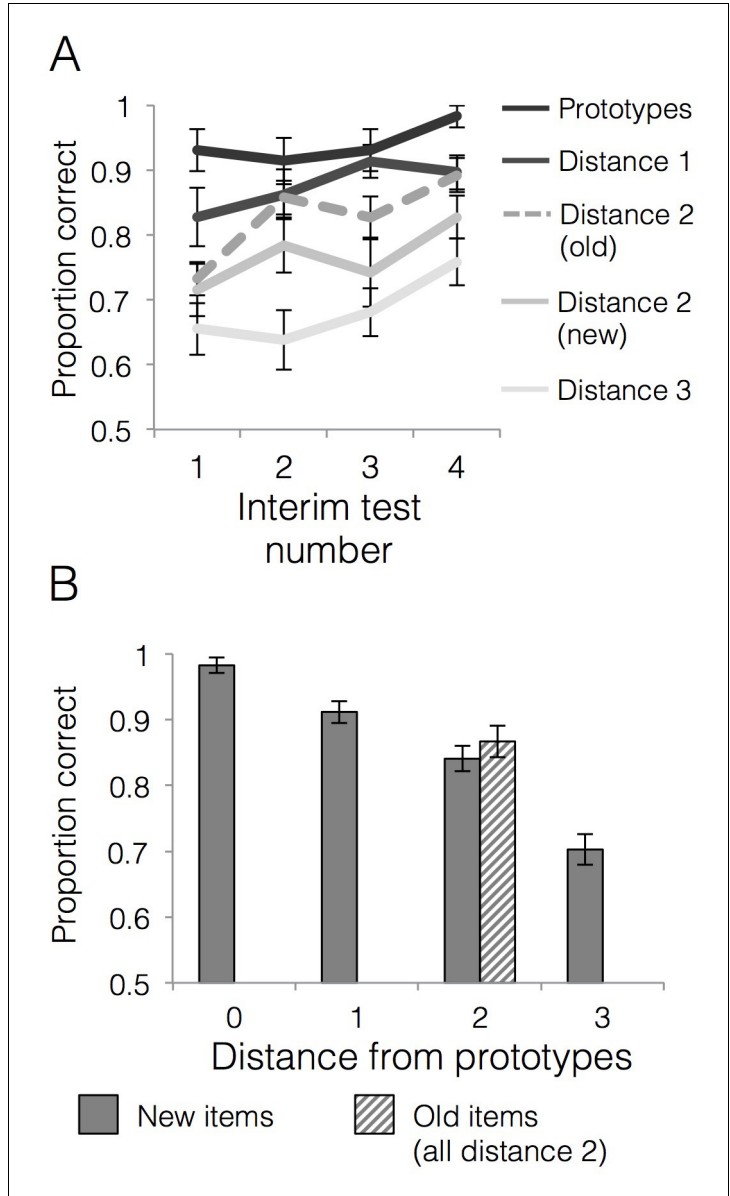

**Figure 3.** Behavioral accuracy for interim and final tests. (**A**) Mean generalization accuracy across each of four interim tests completed during the learning phase. Source data can be found in *Figure 3—source data 1*. (**B**) Mean categorization accuracy in the final test. Source data can be found in *Figure 3—source data 2*. In both cases, accuracies are separated by distance from category prototypes (0–3) and old vs. new (applicable to distance two items only). Error bars represent the standard error of the mean.

The online version of this article includes the following source data for figure 3:

**Source data 1.** Behavioral accuracy - interim tests.
**Source data 2.** Behavioral accuracy - final test.

## Behavioral model fits

*Figure 4a-c* presents model fits in terms of raw negative log likelihood for each phase (lower numbers mean lower model fit error and thus better fit). Fits from the two models tend to be correlated. If a subject randomly guesses on the majority trials (such as early in learning), neither model will fit the subject's responses well and the subject will have higher (mis)fit values for both models. As a subject learns and does better on the task, fits of both models will tend to improve because items

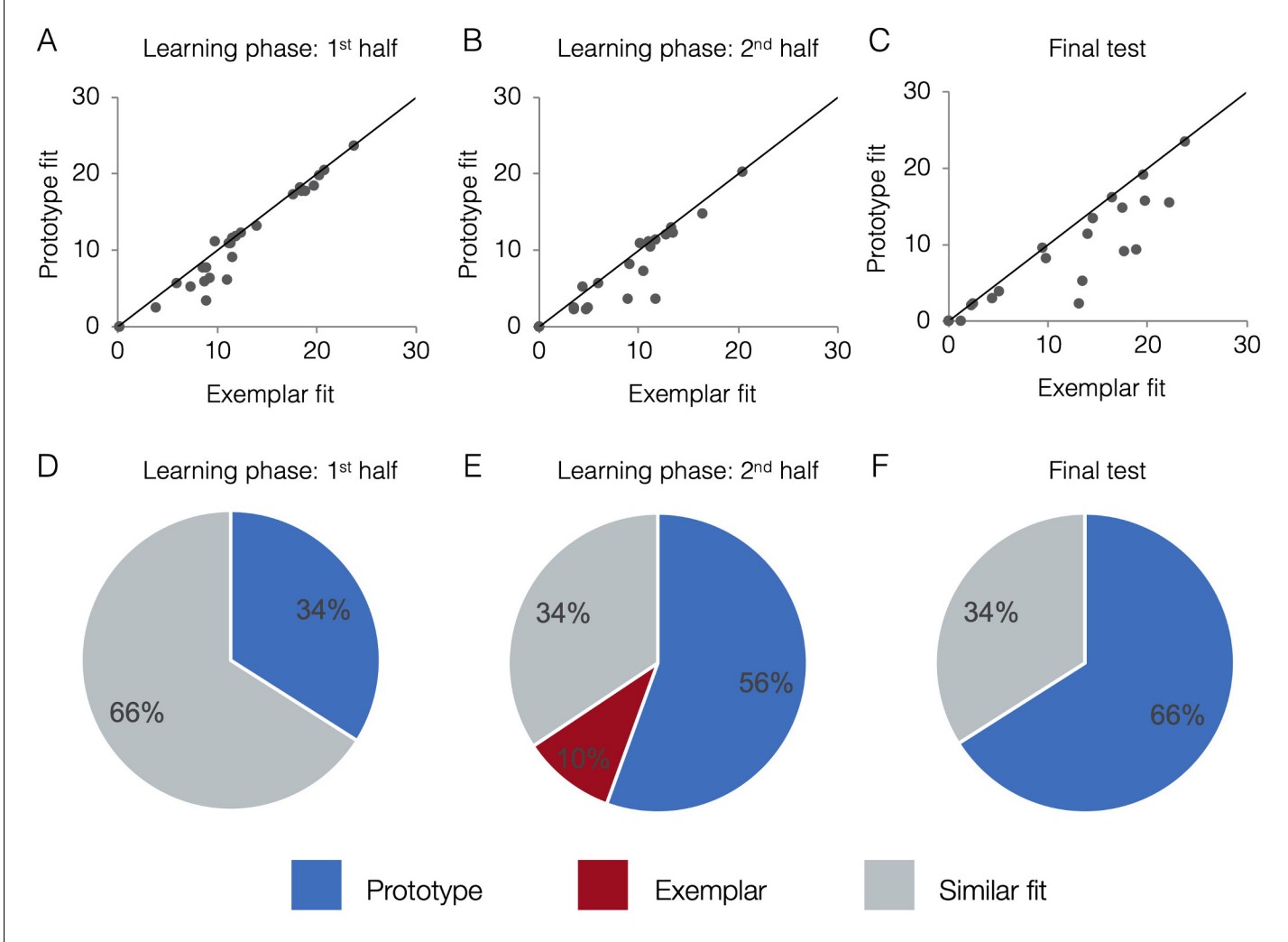

**Figure 4.** Behavioral model fits. Scatter plots indicate the relative exemplar vs. prototype model fits for each subject. Fits are given in terms of negative log likelihood (i.e., model error) such that lower values reflect better model fit. Each dot represents a single subject and the trendline represents equal prototype and exemplar fit. Dots above the line have better exemplar relative to prototype model fit. Dots below the line have better prototype relative to exemplar model fit. Pie charts indicate the percentage of individual subjects classified as best fit by the prototype model (in blue), the exemplar model (in red), and those similarly fit by the two models (in grey). Model fits were computed separately for the 1$^{st}$ half of the learning phase (interim tests 1–2, **A,D**), the 2$^{nd}$ half of the learning phase (interim tests 3–4, **B,E**), and the final test (**C,F**). Source data for all phases can be found in *Figure 4—source data 1*.

The online version of this article includes the following source data for figure 4:

**Source data 1.** Behavioral model fits - all phases.

close to the old exemplars of category A tend to be, on average, closer to the category A prototype than the category B prototype and vice versa. For example, even if a subject had a purely exemplar representation, the prototype model would still fit that subject's behavior quite well, albeit not as well as the exemplar model. Due to the correlation between model fits, the exact fit value for one model is not sufficient to determine a subject's strategy, only the relative fit of one model compared to the other. Visually, in *Figure 4a–c*, subjects above the diagonal are better fit by the exemplar model, participants below the line are better fit by the prototype model, and participants near the line are fit comparably well by both models. Thus, although the model fits tend to be correlated across-subject, the within-subject advantage for one model over another is still detectable and meaningful. To quantify which model fits are comparable and which are reliably different, we took a Monte Carlo approach and compared the observed model fit differences to a null distribution

expected by chance alone (see Methods for details). *Figure 4d–f* presents the percentage of subjects that were classified as having used a prototype strategy, exemplar strategy, or having model fits that were not reliably different from one another ('similar' fit). In the first half of learning, the majority of subjects (66%) had similar prototype and exemplar model fits. In the second half of learning and the final test, the majority of subjects (56% and 66%, respectively) were best fit by the prototype model. Prototype and exemplar model fits may not differ reliably for a given subject, such as when the subject's responses are perfectly consistently with both models (as can happen in high-performing subjects) or when some responses are more consistent with one model while other response are more consistent with the other model. In such cases, a subject may be relying on a single representation but we cannot discern which, or the subject may rely to some extent on both types of representations.

We formally compared model fits for interim tests across the first and second half of the learning phase using a repeated-measures ANOVA on raw model fits. There was a significant main effect of learning phase [$F(1,28)$=39.74, p<0.001, $\eta_p^2 = 0.59$] with better model fits (i.e., lower error) in the second half of the learning phase (M = 5.98, SD = 5.81) compared to the first half (M = 10.64, SD = 6.72). There was also a significant main effect of model [$F(1,28)$=17.50, p<0.001, $\eta_p^2 = 0.39$] with better fit for the prototype model (M = 7.86, SD = 5.95) compared to the exemplar model (M = 8.77, SD = 6.02). The learning phase x model interaction effect was not significant [$F(1,28)$ =0.01, p=0.91, $\eta_p^2 = 0.001$], with a similar prototype advantage in the first half (d = 0.13, $CI_{95}$[0.31,1.45]) as in the second half (d = 0.16, $CI_{95}$[0.22,1.65]). When we compared prototype and exemplar model fits in the final test, we again found a significant advantage for the prototype model over the exemplar model [$t(28)$=3.53, p=0.001, $CI_{95}$[0.89, 3.39], d = 0.23]. Thus, the prototype model provided an overall better fit to behavioral responses throughout the learning phase and final test, and the effect size of the prototype advantage was largest in the final test.

## fMRI
### Model-based MRI
The behavioral model fitting described above maximizes the correspondence between response probabilities generated by the two models and the actual participants' patterns of responses. Once the parameters for the best fitting prototype and best fitting exemplar representations were estimated from the behavioral data, we utilized them to construct model-based fMRI predictors, one exemplar-based predictor and one prototype-based predictor for each participant. For each test item, a model prediction was computed as the similarity of the item to the underlying prototype or exemplar representation regardless of category (representational match; see Methods for details). The trial-by-trial model predictions from both models were then used for fMRI analysis to identify regions that have signal consistent with either model. Importantly, even when behavioral fits are comparable between the two models, the neural model predictions can remain dissociable as they more directly index the underlying representations that are different between the models (*Mack et al., 2013*). For example, the prototypes would be classified into their respective categories with high probability by either model because they are much closer to one category's representation than the other, generating similar behavioral prediction for that trial. However, the representational match will be much higher for the prototype model than the exemplar model as the prototype is not particularly close to any old exemplars. Thus, the neural predictors can dissociate the models to a greater degree than behavioral predictions (*Mack et al., 2013*). Furthermore, the neural model fits can help detect evidence of both kinds of representations, even if one dominates the behavior.

### Learning phase
We first tested the degree to which prototype and exemplar information was represented across ROIs and across different points of the learning phase. Using the data from the interim generalization tests, we compared neural model fits across our six ROIs across the first and second half of the learning phase. Full ANOVA results are presented in *Table 1*. *Figure 5* presents neural model fits for each ROI. *Figure 5A* represents 1$^{st}$ half of the learning phase, *Figure 5B* represents the 2$^{nd}$ half of the learning phase, and *Figure 5C* represents fits collapsed across the entire learning phase (to illustrate the main effects of ROI, model and ROI x model interaction).

**Table 1.** ANOVA results for model-based fMRI during the learning phase.

| Effect | df | F | P | $\eta_p^2$ |
|---|---|---|---|---|
| ROI | 3.4,95.6 GG | 3.90 | .002 | .12 |
| Model | 1,28 | 2.60 | .12 | .09 |
| Learning half | 1,28 | 2.18 | .15 | .07 |
| ROI x Model | 2.9,80.3 GG | 5.91 | .001 | .17 |
| ROI x Learning half | 3.1,86.9 GG | 0.53 | .67 | .02 |
| Model x Learning half | 1,28 | 0.09 | .76 | .003 |
| ROI x Model x Learning Half | 3.2,89.6 GG | 2.31 | .08 | .08 |

As predicted, there was a significant ROI x Model interaction effect, indicating that there were differences across regions in the type of category information that they tracked. To understand the nature of this interaction, we computed follow-up t-tests on the neural model fits in each ROI, collapsed across the first and second half of the learning phase. Consistent with prior work (*Bowman and Zeithamova, 2018*), the VMPFC and anterior hippocampus (our predicted prototype regions) significantly tracked prototype information [VMPFC: $t(28)$ = 2.86, p=0.004, $CI_{95}[\mu >0.06]$, $d$ = 0.75]; [anterior hippocampus: $t(28)$ = 1.88, p=0.04, $CI_{95}[\mu >0.009]$, $d$ = 0.49]. Prototype correlates were numerically but not significantly stronger than exemplar correlates in both regions [VMPFC: $t(28)$ = 1.23, p=0.11, $d$ = 0.34, $CI_{95}[\mu >-0.03]$]; (anterior hippocampus: $t(28)$ = 0.87, p=0.19, $d$ = 0.22, $CI_{95}[\mu >-0.05]$). For the predicted exemplar regions, we found that both lateral parietal cortex and inferior frontal gyrus significantly tracked exemplar model predictions [lateral parietal: $t(28)$ = 2.06, p=0.02, $CI_{95}[\mu >0.02]$, $d$ = 0.54]; [inferior frontal: $t(28)$ = 2.40, p=0.01, $CI_{95}[\mu >0.03]$, $d$ = 0.63], with numerically positive exemplar correlates in lateral occipital cortex that were not statistically significant [$t(28)$=0.78, p=0.22, $CI_{95}[\mu >-0.05]$, $d$ = 0.20]. When comparing neural exemplar fits to neural prototype fits, there was a significant exemplar advantage in both lateral parietal cortex [$t(28)$=3.00, p=0.003, $d$ = 0.71, $CI_{95}[\mu >0.09]$], and in inferior frontal gyrus [$t(28)$=2.63, p=0.01, $d$ = 0.67, $CI_{95}[\mu >0.06]$], that did not reach significance in the lateral occipital cortex [$t(28)$ =1.44, p=0.08, $d$ = 0.36, $CI_{95}[\mu >-0.06]$].

As in our prior study, the posterior hippocampus showed numerically better fit of the exemplar predictor, but neither the exemplar effect [$t(28)$=1.88, p=0.07, $CI_{95}[-0.01,.13]$, $d$ = 0.49] nor the prototype effect reached significance [$t(28)$=−1.14, p=0.26, $CI_{95}[-0.12,.03]$, $d$ = 0.30]. Comparing the effects in the two hippocampal regions as part of a 2 (hippocampal ROI: anterior, posterior) x 2 (model: prototype, exemplar) repeated-measures ANOVA, we found a significant interaction [$F(1,28)$ =9.04, p=0.006, $\eta_p^2$ = 0.24], showing that there is a dissociation along the hippocampal long axis in the type of category information represented. Taken together, we found evidence for different types of category information represented across distinct regions of the brain.

We were also interested in whether there was a shift in representations that could be detected across learning. The only effect that included learning phase that approached significance was the three-way ROI x model x learning phase interaction, likely reflecting the more pronounced region x model differences later in learning (*Figure 5A* vs. *Figure 5B*).

## Final test

*Figure 5D* presents neural model fits from each ROI during the final test. We tested whether the differences across ROIs identified during the learning phase were also present in the final test. As during the learning phase, we found a significant main effect of ROI [$F(2.9,79.8)$=9.13, p<0.001, $\eta_p^2$ = 0.25, GG] and no main effect of model [$F(1,28)$=1.65, p=0.21, $\eta_p^2$ = 0.06]. However, unlike the learning phase, we did not find a significant model x ROI interaction effect [$F(3.3,91.2)$=1.81, p=0.15, $\eta_p^2$ = 0.06, GG]. Because this was a surprising finding, we wanted to better understand what had changed from the learning phase to the final test. Thus, although the ROI x model interaction was not significant in the final test, we computed follow-up tests on regions that had significantly tracked prototype and exemplar predictors during the learning phase. As in the learning phase, both the VMPFC and anterior hippocampus continued to significantly track prototype predictors

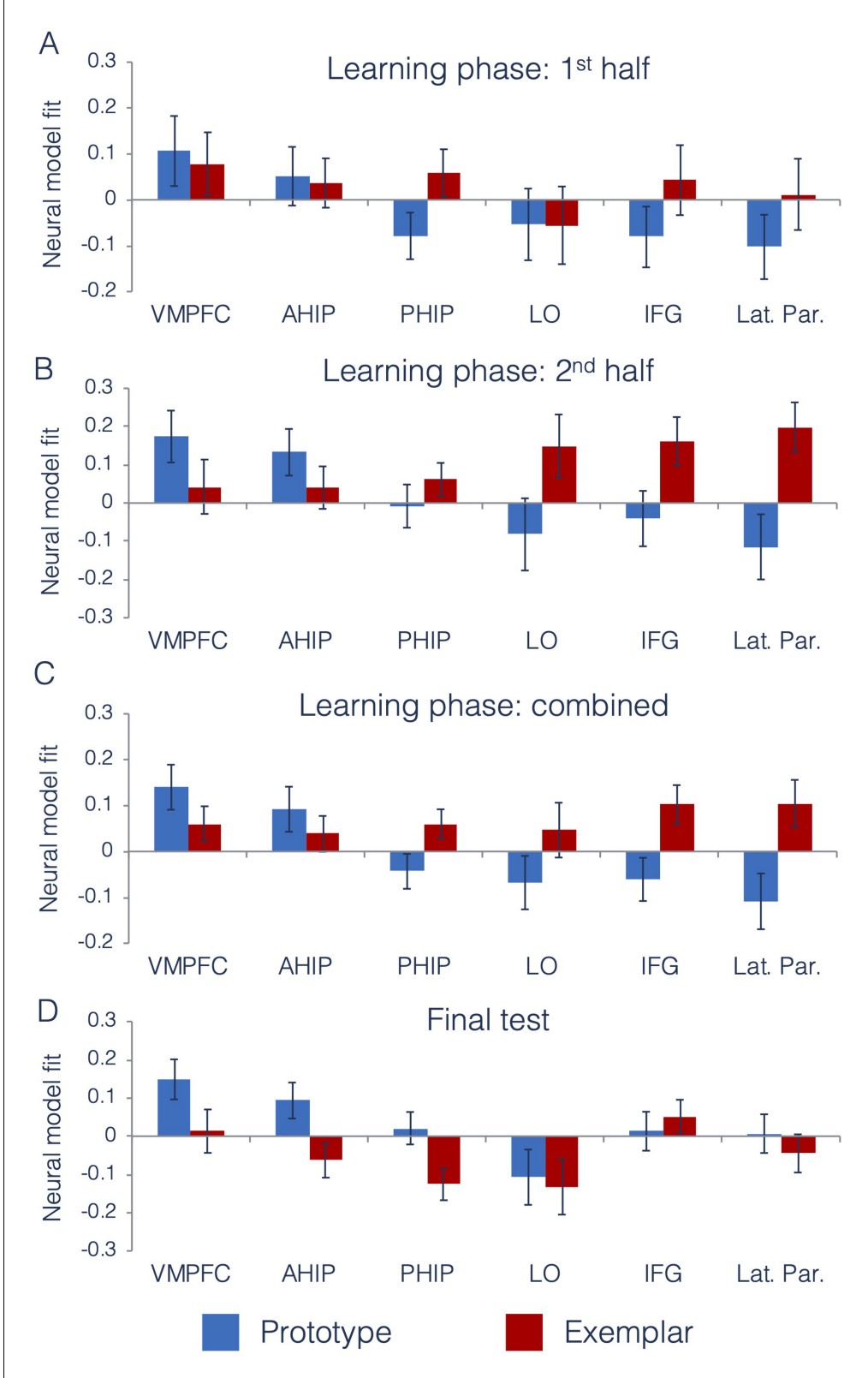

**Figure 5.** Neural prototype and exemplar model fits. Neural model fits for each region of interest for (**A**) the first half of the learning phase, (**B**) the second half of the learning phase, (**C**) the overall learning phase (averaged across the first and second half of learning), and (**D**) the final test. Prototype fits are in blue, exemplar fits in red. Neural model fit is the effect size: the mean/SD of ß-values within each ROI, averaged across appropriate runs. VMPFC = ventromedial prefrontal cortex, ahip = anterior hippocampus, phip = posterior hippocampus, LO = lateral occipital cortex, IFG = inferior
*Figure 5 continued on next page*

*Figure 5 continued*
frontal gyrus, and Lat. Par. = lateral parietal cortex. Source data for interim tests is in *Figure 5—source data 1* and *Figure 5—source data 2* for the final test.
The online version of this article includes the following source data for figure 5:

**Source data 1.** Neural model fits - interim tests.
**Source data 2.** Neural model fits - final test.

during the final test with effect sizes similar to those observed during learning [VMPFC: $t(28) = 2.83$, p=0.004, $CI_{95}[\mu > 0.06]$, $d = 0.74$]; [anterior hippocampus: $t(28) = 1.98$, p=0.03, $CI_{95}[\mu > 0.01]$, $d = 0.52$]. Here, prototype correlates were significantly stronger than exemplar correlates in the anterior hippocampus [$t(28)=2.28$, p=0.02, $d = 0.63$], $CI_{95}[\mu > 0.02]$ and marginally stronger in the VMPFC [$t(28)=1.67$, p=0.053, $d = 0.46$, $CI_{95}[\mu > - 0.03]$]. However, exemplar correlates did not reach significance in any of the predicted exemplar regions (all t < 1.18, p>0.12, d < 0.31).

## Discussion

In the present study, we tested whether exemplar- and prototype-based category representations could co-exist in the brain within a single task under conditions that favor both exemplar memory and prototype extraction. We found signatures of both types of representations across distinct brain regions when participants categorized items during the learning phase. Consistent with predictions based on prior studies, the ventromedial prefrontal cortex and anterior hippocampus tracked abstract prototype information, and the inferior frontal gyrus and lateral parietal cortex tracked specific exemplar information. In addition, we tested whether individuals relied on different types of representations over the course of learning. We did not find evidence of representational shifts either from specific to abstract or vice versa. Instead, results suggested that both types of representations emerged together during learning, although prototype correlates came to dominate by the final test. Together, we show that specific and abstract representations may instead exist in parallel for the same categories.

A great deal of prior work in the domain of category learning has focused on whether classification of novel category members relies on retrieval of individual category exemplars (*Kruschke, 1992*; *Medin and Schaffer, 1978*; *Nosofsky, 1986*; *Nosofsky and Stanton, 2005*; *Zaki et al., 2003*) or instead on abstract category prototypes (*Dubé, 2019*; *Homa, 1973*; *Posner and Keele, 1968*; *Reed, 1972*; *Smith and Minda, 2002*). These two representations are often pitted against one another with one declared the winner over the other, which is based largely on typical model-fitting procedures for behavioral data. Indeed, fitting exemplar and prototype models to behavioral data in the present study generally showed better fit of the prototype model over the exemplar model. However, using neuroimaging allowed us to detect both types of representations apparent across different parts of the brain. These results thus contribute to the ongoing debate about the nature of category representations in behavioral studies of categorization by showing that individuals may maintain multiple representations simultaneously even when one model shows better overall fit to behavior.

In addition to contributing novel findings to a longstanding debate in the behavioral literature, the present study also helps to resolve between prior neuroimaging studies fitting prototype and exemplar models to brain data. Specifically, two prior studies found conflicting results: one study found only exemplar representations in the brain (*Mack et al., 2013*) whereas another found only prototype representations (*Bowman and Zeithamova, 2018*). Notably the brain regions tracking exemplar predictions were different than those identified as tracking prototype predictions, showing that these studies engaged different brain systems in addition to implicating different categorization strategies. However, because the category structures, stimuli and analysis details also differed between these studies, the between-studies differences in the identified neural systems could not be uniquely attributed to the distinct category representations that participants presumably relied on. The present data newly show that neural prototype and exemplar correlates can exist not only across different task contexts but also within the same task, providing evidence that these neural differences reflect distinct category representations rather than different task details.

Moreover, our results aligned with those found separately across two studies, replicating the role of the VMPFC and anterior hippocampus in tracking prototype information (*Bowman and Zeithamova, 2018*) and replicating the role of inferior prefrontal and lateral parietal cortices in tracking exemplar information (*Mack et al., 2013*). Prior work has shown that the hippocampus and VMPFC support integration across related experiences in episodic inference tasks (for reviews, see *Schlichting and Preston, 2017*; *Zeithamova and Bowman, 2020*). We have now shown for the second time that these same regions also track prototype information during category generalization, suggesting that they may play a common role across seemingly distinct tasks. That is, integrating across experiences may not only link related elements as in episodic inference tasks, but may also serve to derive abstract information such as category prototypes. We also replicated a dissociation within the hippocampus from *Bowman and Zeithamova, 2018* in which the anterior hippocampus showed significantly stronger prototype representations than the posterior hippocampus. Our findings are consistent with a proposed gradient along the hippocampal long axis, with representations becoming increasingly coarse in spatial and temporal scale moving from posterior to anterior portions of the hippocampus (*Brunec et al., 2018*; *Poppenk et al., 2013*). Lastly, we note that while the VMPFC significantly tracked prototype predictions, there was only a marginal difference between prototype and exemplar correlates in this region. Thus, it remains an open question whether representations in VMPFC are prototype specific or instead may reflect some mix of coding.

Our finding that IFG and lateral parietal cortices tracked exemplar predictions is consistent not only with prior work showing exemplar correlates in these regions during categorization (*Mack et al., 2013*), but also with the larger literature on their role in maintaining memory specificity. In particular, IFG is thought to play a critical role in resolving interference between similar items (*Badre and Wagner, 2005*; *Bowman and Dennis, 2016*; *Jonides et al., 1998*; *Kuhl et al., 2007*) while lateral parietal cortices often show high fidelity representations of individual items and features necessary for task performance (*Kuhl and Chun, 2014*; *Xiao et al., 2017*). The present findings support and further this prior work by showing that regions supporting memory specificity across many memory tasks may also contribute to exemplar-based concept learning.

In addition to IFG and lateral parietal cortex, we predicted that lateral occipital cortex would track exemplar information. This prediction was based both on its previously demonstrated exemplar correlates in the Mack et al. study as well as evidence that representations in visual regions shift with category learning (*Folstein et al., 2013*; *Freedman et al., 2001*; *Myers and Swan, 2012*; *Palmeri and Gauthier, 2004*). Such shifts are posited to be the result of selective attention to visual features most relevant for categorization (*Goldstone and Steyvers, 2001*; *Medin and Schaffer, 1978*; *Nosofsky, 1986*). Consistent with the selective attention interpretation, Mack et al., showed that LO tracked similarity between items when feature weights estimated by the exemplar model were taken into account, above-and-beyond tracking physical similarity. In the present study, LO showed an overall similar pattern as IFG and lateral parietal cortex, but exemplar correlates did not reach significance during any phase of the experiment, providing only weak evidence for exemplar coding in this region. However, in contrast to this prior work, all stimulus features in our study were equally relevant for determining category membership. This aspect of our task may have limited the role of selective attention in the present study and thus the degree to which perceptual regions tracked category information.

In designing the present study, we aimed to increase exemplar strategy use as compared to our prior study in which the prototype model fit reliably better than the exemplar model in 73% of the sample (*Bowman and Zeithamova, 2018*). We included a relatively coherent category structure that was likely to promote prototype formation (*Bowman and Zeithamova, 2018*; *Bowman and Zeithamova, 2020*), but tried to balance it with an observational rather than feedback-based training task in hopes of emphasizing individual items and promoting exemplar representations. The results suggest some shift in model fits, albeit modest. The prototype strategy was still identified as dominant in the latter half of learning and the final test, but we also observed more participants who were comparably fit by both models. Moreover, we detected exemplar correlates in the brain in the present study, albeit only during the second half of the learning phase. Thus, while the behavioral shift in model fits was modest, it may have been sufficient to make exemplar representations detectable despite prototype dominance in behavior. Notably, our prior study did show some evidence of exemplar-tracking regions (including portions of LO and lateral parietal cortex) but only when we

used a lenient, uncorrected threshold. This suggests that exemplar-based representations may form in the brain even though they are not immediately relevant for the task at hand.

It may be that representations form at multiple levels of specificity to promote flexibility in future decision-making because it is not always clear what aspects of current experience will become relevant (*Zeithamova and Bowman, 2020*). Consistent with this idea, research shows that category representations can spontaneously form alongside memory for individuals, even when task instructions emphasize distinguishing between similar individuals (*Ashby et al., 2020*). In the present context, accessing prototype representations may be efficient for making generalization judgments, but they cannot on their own support judgments that require discrimination between separate experiences or between members of the same category. Thus, exemplar representations may also form to support judgments requiring specificity. Precedence for co-existing representations also comes from neuroimaging studies of spatial processing (*Brunec et al., 2018*), episodic inference (*Schlichting et al., 2015*), and memory for narratives (*Collin et al., 2015*). These studies have all shown evidence for separate representations of individual experiences alongside representations that integrate across experiences. The present results show that these parallel representations may also be present during category learning.

While co-existing prototype and exemplar representations were clear during the learning phase of our task, they were not present during the final test phase. The VMPFC and anterior hippocampus continued to track prototypes during the final test, but exemplar-tracking regions no longer emerged. The lack of exemplar correlates in the brain was matched by a weaker exemplar effect in behavior. While we observed a reliable advantage for old relative to new items matched for distance during the interim tests, the old advantage was no longer significant in the final test. The effect size for the prototype advantage in model fits was also larger in the final test than in the learning phase. This finding was unexpected, but we offer several possibilities that can be investigated further in future research. One possibility is that exemplar representations were weakened in the absence of further observational study runs that had boosted exemplars in earlier phases. Similarly, framing it as a 'final test' may have switched participants from trying to gather multiple kinds of information that might improve later performance (i.e., both exemplar and prototype) to simply deploying the strongest representation that they had, which seems to have been prototype-based. Alternatively, there may be real, non-linear dynamics in how prototype and exemplar representations develop. For example, exemplar representations may increase up to some threshold while individuals are encoding these complex stimuli, then decrease as a result of repetition suppression (*Desimone, 1996*; *Gonsalves et al., 2005*; *Henson et al., 2002*) once individual items are sufficiently well represented. Of course, future studies will be needed to both replicate this finding and directly test these differing possibilities.

In addition to identifying multiple, co-existing types of category representations during learning, we sought to test whether there were representational shifts as category knowledge developed. While there was a prototype advantage in the brain during the final test, we found no evidence for a shift between exemplar and prototype representations over the course of learning. Both prototype and exemplar correlates showed numerical increases across learning in brain and behavior, suggesting strengthening of both types of representations in parallel. Prior work has shown that individuals may use both rules and memory for individual exemplars throughout learning without strong shifts from one to the other (*Thibaut et al., 2018*). Others have suggested that there may be representational shifts during category learning, but rather than shifting between exemplar and prototype representations, early learning may be focused on detecting simple rules and testing multiple hypotheses (*Johansen and Palmeri, 2002*; *Nosofsky et al., 1994*; *Paniukov and Davis, 2018*), whereas similarity-based representations such as prototype and exemplar representations may develop later in learning (*Johansen and Palmeri, 2002*). Our findings are consistent with this framework, with strong prototype and exemplar representations emerging across distinct regions primarily in the second half of learning. Our results are also consistent with recent neuroimaging studies showing multiple memory representations forming in parallel without need for competition (*Collin et al., 2015*; *Schlichting et al., 2015*), potentially allowing individuals to flexibly use prior experience based on current decision-making demands.

## Conclusion

In the present study, we found initial evidence that multiple types of category representations may co-exist across distinct brain regions within the same categorization task. The regions identified as prototype-tracking (anterior hippocampus and VMPFC) and exemplar-tracking (IFG and lateral parietal cortex) in the present study align with prior studies that have found only one or the other. These findings shed light on the multiple memory systems that contribute to concept representation and provide novel evidence of how the brain may flexibly represent information at different levels of specificity and that these representations may not always compete during learning.

# Materials and methods

## Participants

Forty volunteers were recruited from the University of Oregon and surrounding community and were financially compensated for their research participation. This sample size was determined based on effect sizes for neural prototype-tracking and exemplar-tracking regions estimated from prior studies (*Bowman and Zeithamova, 2018*; *Mack et al., 2013*), allowing for detection of the minimum effect size (prototype correlates in anterior hippocampus, $d = 0.43$ with $n = 29$) using a one-tailed, one-sample t-test with at least 80% power. All participants provided written informed consent, and Research Compliance Services at the University of Oregon approved all experimental procedures. All participants were right-handed, native English speakers and were screened for neurological conditions, medications known to affect brain function, and contraindications for MRI.

A total of 11 subjects were excluded. Six subjects were excluded prior to fMRI analyses: three subjects for chance performance (<0.6 by the end of the training phase and/or <0.6 for trained items in the final test), one subject for excessive motion (>1.5 mm within multiple runs), and two subjects for failure to complete all phases. An additional five subjects were excluded for high correlation between fMRI regressors that precluded model-based fMRI analyses of the first or second half of learning phase: three subjects had $r > 0.9$ for prototype and exemplar regressors and two subjects had a rank deficient design matrix driven by a lack of trial-by-trial variability in the exemplar predictor. In all five participants, attentional weight parameter estimates from both models indicated that most stimulus dimensions were ignored, which in some cases may lead to a lack of variability in model fits. This left 29 subjects (age: $M = 21.9$ years, $SD = 3.3$ years, range 18–30 years; 19 females) reported in all analyses. Additionally, we excluded single runs from three subjects who had excessive motion limited to that single run.

## Materials

Stimuli consisted of cartoon animals that differed on eight binary features: neck (short vs. long), tail (straight vs. curled), foot shape (claws vs. round), snout (rounded vs. pig), head (ears vs. antennae), color (purple vs. red), body shape (angled vs. round), and design on the body (polka dots vs. stripes) (*Bozoki et al., 2006*; *Zeithamova et al., 2008*; available for download osf.io/8bph2). The two possible versions of all features can be seen across the two prototypes shown in *Figure 1C*. For each participant, the stimulus that served as the prototype of category A was randomly selected from four possible stimuli and all other stimuli were re-coded in reference to that prototype. The stimulus that shared no features with the category A prototype served as the category B prototype. Physical distance between any pair of stimuli was defined by their number of differing features. Category A stimuli were those that shared more features with the category A prototype than the category B prototype. Category B stimuli were those that shared more features with the category B prototype than the category A prototype. Stimuli equidistant from the two prototypes were not used in the study.

### Training set

The training set included four stimuli per category, each differing from their category prototype by two features (see *Table 2* for training set structure). The general structure of the training set with regard to the category prototypes was the same across subjects, but the exact stimuli differed based on the prototypes selected for a given participant. The training set structure was selected to generate many pairs of training items that were four features apart both within the same category and

**Table 2.** Dimension values for example prototypes and training stimuli from each category.

| Stimulus | Dimension values | | | | | | | |
| --- | --- | --- | --- | --- | --- | --- | --- | --- |
| | 1 | 2 | 3 | 4 | 5 | 6 | 7 | 8 |
| Prototype A | 1 | 1 | 1 | 1 | 1 | 1 | 1 | 1 |
| A1 | 1 | 1 | 1 | 1 | 1 | 1 | 0 | 0 |
| A2 | 0 | 1 | 1 | 1 | 0 | 1 | 1 | 1 |
| A3 | 1 | 0 | 1 | 0 | 1 | 1 | 1 | 1 |
| A4 | 1 | 1 | 0 | 1 | 1 | 0 | 1 | 1 |
| Prototype B | 0 | 0 | 0 | 0 | 0 | 0 | 0 | 0 |
| B1 | 0 | 0 | 0 | 0 | 0 | 0 | 1 | 1 |
| B2 | 1 | 0 | 0 | 0 | 1 | 0 | 0 | 0 |
| B3 | 0 | 1 | 0 | 1 | 0 | 0 | 0 | 0 |
| B4 | 0 | 0 | 1 | 0 | 0 | 1 | 0 | 0 |

across the two categories. This design ensured that categories could not be learned via unsupervised clustering based on similarity of exemplars alone.

## Interim test sets

Stimuli in the interim generalization tests included 22 unique stimuli: the eight training stimuli, both prototypes, and two new stimuli at each distance (1, 2, 3, 5, 6, 7) from the category A prototype. Distance 1, 2 and 3 items were scored as correct when participant labeled them as category A members. Items at the distance 5, 6 and 7 from the category A prototype (thus distance 3, 2, and one from the B prototype) were scored as correct when participant labeled them as category B members. While new unique distance 1–3, 5–7 items were selected for each interim test set, the old training stimuli and the prototypes were necessarily the same for each test.

## Final test set

Stimuli in the final test included 58 unique stimuli. Forty-eight of those consisted of 8 new stimuli selected at each distance 1–3, 5–7 from the category A prototype, each presented once during the final test. These new items were distinct from those used in either the training set or the interim test sets with the exception of the items that differed by only one feature from their respective prototypes. Because there are only 8 distance one items for each prototype, they were all used as part of the interim test sets before being used again in the final test set. The final test also included the eight training stimuli and the two prototypes, each presented twice in this phase (*Bowman and Zeithamova, 2018*; *Kéri et al., 2001*; *Smith et al., 2008*). The stimulus structure enabled dissociable behavioral predictions from the two models. While stimuli near the prototypes also tend to be near old exemplars, the correlation is imperfect. For example, when attention is equally distributed across features, the prototype model would make the same response probability prediction for all distance three items. However, some of those distance three items were near an old exemplar while others were farther from all old exemplars, creating distinct exemplar model predictions. Because we varied the test stimuli to include all distances from the prototypes, and because within each distance to the prototype there was variability in how far the stimuli are from the old exemplars, the structure was set up to facilitate dissociation between the model predictions.

## Experimental design

The study consisted of two sessions: one session of neuropsychological testing and one experimental session. Only results from the experimental session are reported in the present manuscript. In the experimental session, subjects underwent four cycles of observational study and interim generalization tests (*Figure 1D*), followed by a final generalization test (*Figure 1E*), all while undergoing fMRI.

In each run of observational study, participants were shown individual animals on the screen with a species label (Febbles and Badoons) and were told to try to figure out what makes some animals Febbles and others Badoons without making any overt responses. Each stimulus was presented on

the screen for 5 s followed by a 7 s ITI. Within each study run, participants viewed the training examples three times in a random order. After two study runs, participants completed an interim generalization test. Participants were shown cartoon animals without their labels and classified them into the two species without feedback. Each test stimulus was presented for 5 s during which time they could make their response, followed by a 7 s ITI. After four study-test cycles, participants completed a final categorization test, split across four runs. As in the interim tests, participants were asked to categorize animals into one of two imaginary species (Febbles and Badoons) using the same button press while the stimulus was on the screen. Following the MRI session, subjects were asked about the strategies they used to learn the categories, if any, and then indicated which version of each feature they thought was most typical for each category. Lastly, subjects were verbally debriefed about the study.

## fMRI Data Acquisition

Raw MRI data are available for download via OpenNeuro (openneuro.org/datasets/ds002813). Scanning was completed on a 3T Siemens MAGNETOM Skyra scanner at the University of Oregon Lewis Center for Neuroimaging using a 32-channel head coil. Head motion was minimized using foam padding. The scanning session started with a localizer scan followed by a standard high-resolution T1-weighted MPRAGE anatomical image (TR 2500 ms; TE 3.43 ms; TI 1100 ms; flip angle 7°; matrix size 256 256; 176 contiguous slices; FOV 256 mm; slice thickness 1 mm; voxel size 1.0 1.0 1.0 mm; GRAPPA factor 2). Then, a custom anatomical T2 coronal image (TR 13,520 ms; TE 88 ms; flip angle 150°; matrix size 512 512; 65 contiguous slices oriented perpendicularly to the main axis of the hippocampus; interleaved acquisition; FOV 220 mm; voxel size 0.4 0.4 2 mm; GRAPPA factor 2) was collected. This was followed by 16 functional runs using a multiband gradient echo pulse sequence [TR 2000 ms; TE 26 ms; flip angle 90°; matrix size 100 100; 72 contiguous slices oriented 15° off the anterior commissure–posterior commissure line to reduced prefrontal signal dropout; interleaved acquisition; FOV 200 mm; voxel size 2.0 2.0 2.0 mm; generalized autocalibrating partially parallel acquisitions (GRAPPA) factor 2]. One hundred and forty-five volumes were collected for each observational study run, 133 volumes for each interim test run, and 103 volumes for each final test run.

## Behavioral accuracies

### Interim tests

To assess changes in generalization accuracy across train-test cycles, we computed a 4 (interim test run: 1–4) x 4 (distance: 0–3) repeated-measures ANOVA on accuracy for new items only. We were particularly interested in linear effects of interim test run and distance. We also tested whether there was a difference across training in accuracy for the training items themselves versus new items at the same distance from their prototypes, which can index how much participants learn about specific items above-and-beyond what would be expected based on their typicality. We thus computed a 4 (interim test run: 1–4) x 2 (item type: training, new) repeated-measures ANOVA on accuracies for distance two items.

### Final test

First, to assess the effect of item typicality, classification performance in the final test (collapsed across runs) was assessed by computing a one-way, repeated-measures ANOVA across new items at distances (0–3) from either prototype. Second, we assessed whether there was an old-item advantage by comparing accuracy for training items and new items of equal distance from prototypes (distance 2) using a paired-samples t-test. For all analyses (including fMRI analyses described below), a Greenhouse-Geisser correction was applied whenever the assumption of sphericity was violated as denoted by 'GG' in the results.

## Prototype and exemplar model fitting

As no responses were made during the study runs, prototype and exemplar models were only fit to test runs – interim and final tests. As the number of trials in each interim test was kept low to minimize exposure to non-training items during the learning phase, we concatenated across interim tests 1 and 2 and across interim tests 3 and 4 to obtain more robust model fit estimates for the first half

vs. second half of the learning phase. Model fits for the final test were computed across all four runs combined. Each model was fit to trial-by-trial data in individual participants.

## Prototype similarity

As in prior studies (*Bowman and Zeithamova, 2018*; *Maddox et al., 2011*; *Minda and Smith, 2001*), the similarity of each test stimulus to each prototype was computed, assuming that perceptual similarity is an exponential decay function of physical similarity (*Shepard, 1957*), and taking into account potential differences in attention to individual features. Formally, similarity between the test stimulus and the prototypes was computed as follows:

$$S_A(x) = exp\left[-c\sum_{i=1}^{m}(w_i|x_i - proto_{Ai}|^r)^{1/r}\right],$$ (1)

where $S_A(x)$ is the similarity of item $x$ to category A, $x_i$ represents the value of the item $x$ on the $i$th dimension of its $m$ binary dimensions ($m$ = 8 in this study), $proto_A$ is the prototype of category A, $r$ is the distance metric (fixed at one for the city-block metric for the binary dimension stimuli). Parameters that were estimated from each participant's pattern of behavioral responses were $w$ (a vector with eight weights, one for each of the eight stimulus features and constrained to sum to 1) and $c$ (sensitivity: the rate at which similarity declines with physical distance, constrained to be 0–100).

## Exemplar similarity

Exemplar models assume that categories are represented by their individual exemplars, and that test items are classified into the category with the highest summed similarity across category exemplars (*Figure 1A*). As in the prototype model, a nonlinear exponential decay function is used to transform physical similarity into subjective similarity, based on research on how perceived similarity relates to physical similarity (*Shepard, 1957*). Using a nonlinear function has the effect of weighting the most similar exemplars more heavily than the least similar exemplars, as the similarity value for two items that are physically one feature apart will be more than twice the similarity value of two items that are two features apart. Using the sum of similarity across all exemplars within a category provides an opportunity for multiple highly similar exemplars to be considered in making decisions, which allows the model to generate different predictions when there is a single close exemplar versus when there are multiple close exemplars. Together, this means that the most similar training exemplars drive the summed similarity metric but there is still differentiation in the predictions informed by other exemplars beyond the closest exemplar. This is canonically how an item's similarity to each category is computed in exemplar models (*Nosofsky, 1987*; *Zaki et al., 2003*).Formally, similarity of each test stimulus to the exemplars of each category was computed as follows:

$$S_A(x) = \sum_{y\in A}exp\left[-c\sum_{i=1}^{m}(w_i|x_i - y_i|^r)^{1/r}\right]$$ (2)

where $y$ represents the individual training stimuli from category A, and the remaining notation and parameters are as in *Equation 1*.

## Parameter estimation

For both models, the probability of assigning a stimulus $x$ to category A is equal to the similarity to category A divided by the summed similarity to categories A and B, formally, as follows:

$$P(A|x) = \frac{S_A(x)}{S_A(x) + S_B(x)}$$ (3)

Using these equations, the best fitting $w_{1-8}$ (attention to each feature) and $c$ (sensitivity) parameters were estimated from the behavioral data of each participant, separately for the first half of the learning phase, second half of the learning phase, and the final test, and separately for the prototype and exemplar models. To estimate these parameters for a given model, the trial-by-trial predictions generated by *Equation 3* were compared with the participant's actual series of responses, and model parameters were tuned to minimize the difference between predicted and observed

responses. An error metric (negative log likelihood of the entire string of responses) was computed for each model by summing the negative of log-transformed probabilities, and this value was minimized by adjusting $w$ and $c$ using standard maximum likelihood methods, implemented in MATLAB (Mathworks, Natick, MA), using the 'fminsearch' function.

## Group analyses

After optimization, we computed a 2 (model: prototype, exemplar) x 2 (learning phase half: 1st, 2nd) repeated-measures ANOVA on the model fit values (i.e., negative log likelihood) to determine which model provided a better fit to behavioral responses at the group level and if there were shifts across learning in which model fit best. We used a paired-samples t-test comparing model fits during the final test to determine whether the group as a whole was better fit by the prototype or exemplar model by the end of the experiment.

We also tested whether individual subjects were reliably better fit by one model or the other using a permutation analysis. For each subject in each phase of the experiment, we created a null distribution of model fits by shuffling the order of stimuli associated with the subject's actual string of responses, then fitting the prototype and exemplar models to this randomized set of response – stimulus mappings. We repeated this process 10,000 times for each subject in each phase. We first confirmed that the actual prototype and exemplar model fits were reliably better than would be expected if subjects were responding randomly by comparing these real fits to the null distribution of prototype and exemplar model fits (alpha = 0.05, one-tailed). Indeed, both the prototype and exemplar models fit reliably better than chance for all subjects in all phases of the experiment. Next, we tested whether one model reliably outperformed the other model by taking the difference in model fits generated by the permutation analysis. We then compared the observed difference in model fits to the null distribution of model fit differences and determined whether the observed difference appeared with a frequency of less than 5% (alpha = 0.05, two-tailed). Using this procedure, we labeled each subject as having used a prototype strategy, exemplar strategy, or having fits that did not differ reliably from one another ('similar' model fits) for each phase of the experiment.

## fMRI Preprocessing

The raw data were converted from dicom files to nifti files using the dcm2niix function from MRIcron (https://www.nitrc.org/projects/mricron). Functional images were skull-stripped using BET (Brain Extraction Tool), which is part of FSL (http://www.fmrib.ox.ac.uk/fsl). Within-run motion correction was computed using MCFLIRT in FSL to realign each volume to the middle volume of the run. Across-run motion correction was then computed using ANTS (Advanced Normalization Tools) by registering the first volume of each run to the first volume of the first functional run (i.e., the first training run). Each computed transformation was then applied to all volumes in the corresponding run. Brain-extracted and motion-corrected images from each run were entered into the FEAT (fMRI Expert Analysis Tool) in FSL for high-pass temporal filtering (100 s) and spatial smoothing (4 mm FWHM kernel).

## Regions of interest

Regions of interest (ROIs, *Figure 2*) were defined anatomically in individual participants' native space using the cortical parcellation and subcortical segmentation from Freesurfer version 6 (https://surfer.nmr.mgh.harvard.edu/) and collapsed across hemispheres to create bilateral masks. Past research has indicated that there may be a functional gradient along the hippocampal long axis, with detailed, find-grained representations in the posterior hippocampus and increasingly coarse, generalized representations proceeding toward the anterior hippocampus (*Brunec et al., 2018*; *Frank et al., 2019*; *Poppenk et al., 2013*). As such, we divided the hippocampal ROI into anterior/posterior portions at the middle slice. When a participant had an odd number of hippocampal slices, the middle slice was assigned to the posterior hippocampus. Based on our prior report (*Bowman and Zeithamova, 2018*), we expected the anterior portion of the hippocampus to track prototype predictors, together with VMPFC (medial orbitofrontal label in Freesurfer). Based on the prior study by *Mack et al., 2013*, we expected lateral occipital cortex, inferior frontal gyrus (combination of pars opercularis, pars orbitalis, and pars triangularis freesurfer labels), and lateral parietal cortex (combination of inferior parietal and superior parietal freesurfer labels) to track exemplar

predictors. The posterior hippocampus was also included as an ROI, to test for an anterior/posterior dissociation within the hippocampus. While one might expect the posterior hippocampus to track exemplar predictors based on the aforementioned functional gradient, our prior report *Bowman and Zeithamova, 2018* found only a numeric trend in this direction and *Mack et al., 2013* did not report any hippocampal findings despite significant exemplar correlates found in the cortex. Thus, we did not have strong predictions regarding the posterior hippocampus, other than being distinct from the anterior hippocampus.

## Model-based fMRI analyses

fMRI data were modeled using the GLM. Three task-based regressors were included in the GLM: one for all trial onsets, one that included modulation for each trial by prototype model predictions, and one that included modulation for each trial by exemplar model predictions. Events were modeled with a duration of 5 s, which was the fixed length of the stimulus presentation. Onsets were then convolved with the canonical hemodynamic response function as implemented in in FSL (a gamma function with a phase of 0 s, and SD of 3 s, and a mean lag time of 6 s). The six standard timepoint-by-timepoint motion parameters were included as regressors of no interest.

The regressor for all trial onsets was included to account for activation that is associated with performing a categorization task generally, but does not track either model specifically. The modulation values for each model were computed as the summed similarity across category A and category B (denominator of *Equation 3*) generated by the assumptions of each model (from *Equations 1 and 2*). This summed similarity metric indexes how similar the current item is to the existing category representations as a whole (regardless of which category it is closer to) and has been used by prior studies to identify regions that contain such category representations (*Bowman and Zeithamova, 2018*; *Davis and Poldrack, 2014*; *Mack et al., 2013*). Correlations between prototype and exemplar summed similarity values ranged from r = −0.73 to. 82 for included subjects, with a mean of absolute values of r = 0.32. A vast majority (80%) of included runs had prototype and exemplar correlations between +/-. 5. To account for any shared variance between the regressors, we included both model predictors in the same GLM. We verified that the pattern of results remained the same when analyses are limited to participants with absolute correlations r < 0.5 in all runs, with most correlations being quite small.

For region of interest analyses, we obtained an estimate of how much the BOLD signal in each region tracked each model predictor by dividing the mean ROI parameter estimate by the standard deviation of parameter estimates (i.e., computing an effect size measure). Normalizing the beta values by their error of the estimate de-weighs values associated with large uncertainty, similar to how lower level estimates are used in group analyses as implemented in FSL (*Smith et al., 2004*). These normalized beta values were then averaged across the appropriate runs (interim tests 1–2, interim tests 3–4, all four runs of the final test) and submitted to group analyses.

We tested whether prototype and exemplar correlates emerged across different regions and/or at different points during the learning phase. To do so, we computed a 2 (model: prototype, exemplar) x 2 (learning phase: 1st half, 2nd half) x 6 (ROI: VMPFC, anterior hippocampus, posterior hippocampus, lateral occipital, lateral prefrontal, and lateral parietal cortices) repeated-measures ANOVA on parameter estimates from the interim test runs. We were interested in a potential model x ROI interaction effect, indicating differences across brain regions in the type of category information represented. Following any significant interaction effect, we computed one-sample t-tests to determine whether each region significantly tracked a given model and paired-samples t-tests to determine whether the region tracked one model reliably better than the other. Given a priori expectations about the nature of these effects, we computed one-tailed tests only on the effects of interest: for example, in hypothesized prototype-tracking ROIs (anterior hippocampus and VMPFC), we computed one-sample t-tests to compare prototype effects to zero and a paired-samples t-test to test whether the prototype correlates were stronger than exemplar correlates. We followed a similar procedure in hypothesized exemplar-tracking ROIs (inferior frontal gyrus, lateral parietal cortex, lateral occipital cortex). We were also interested in potential interactions with the learning phase, which would indicate shift across learning in category representations. Following any such interaction, follow-up ANOVAs or t-tests were performed to better understand drivers of the effect.

We next tested ROI differences in the final generalization phase. To do so, we computed a 2 (model: prototype, exemplar) x 6 (ROI: see above) repeated-measures ANOVA on parameter

estimates from the final generalization test. We were particularly interested in the model x ROI interaction effect, which would indicate that regions differ in which model they tracked. Because each participant's neural model fit is inherently dependent on their behavioral model fit, we focused on group-average analyses and did not perform any brain-behavior individual differences analyses.

## Acknowledgements

Funding for this work was provided in part by the National Institute of Neurological Disorders and Stroke Grant R01-NS112366 (DZ), National Institute on Aging Grant F32-AG054204 (CRB), and Lewis Family Endowment, which supports the Robert and Beverly Lewis Center for Neuroimaging at the University of Oregon (DZ).

## Additional information

### Funding

| Funder | Grant reference number | Author |
| --- | --- | --- |
| National Institute on Aging | F32-AG-054204 | Caitlin R Bowman |
| National Institute of Neurological Disorders and Stroke | R01-NS112366 | Dasa Zeithamova |
| University of Oregon | Robert and Beverly Lewis Center for Neuroimaging | Dagmar Zeithamova |

The funders had no role in study design, data collection and interpretation, or the decision to submit the work for publication.

### Author contributions

Caitlin R Bowman, Conceptualization, Resources, Data curation, Software, Formal analysis, Supervision, Validation, Investigation, Visualization, Methodology, Writing - original draft, Project administration; Takako Iwashita, Resources, Investigation, Project administration, Writing - review and editing; Dagmar Zeithamova, Conceptualization, Resources, Supervision, Funding acquisition, Methodology, Writing - review and editing

### Author ORCIDs

Caitlin R Bowman https://orcid.org/0000-0002-5833-3591

### Ethics

Human subjects: All participants provided written informed consent, and Research Compliance Services at the University of Oregon approved all experimental procedures (approval code 10162014.010).

### Decision letter and Author response

Decision letter https://doi.org/10.7554/eLife.59360.sa1
Author response https://doi.org/10.7554/eLife.59360.sa2

## Additional files

### Supplementary files

• Transparent reporting form

### Data availability

Raw MRI data have been deposited at openneuro.org/datasets/ds002813. Source data have been provided for Figures 3-5.

The following dataset was generated:

| Author(s) | Year | Dataset title | Dataset URL | Database and Identifier |
|---|---|---|---|---|
| Bowman CR, Iwashita T, Zeithamova D | 2020 | Model-based fMRI reveals co-existing specific and generalized concept representations | https://openneuro.org/datasets/ds002813 | OpenNeuro, ds002813 |

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
