## [Decision Letter]

**Acceptance summary:**

The ability to form categories is critical to organizing our knowledge about the world. However, the nature by categories are formed has been the subject of longstanding debate, with the field divided between two competing ideas. One class of theories holds that categories are represented in terms of their individual members ("exemplars"), whereas another class of theories states that categories are represented by the abstracted average of the most typical members of the category ("prototypes"). Little progress had been made on this debate, until Bowman and colleagues show here that both exemplar and prototype representations exist simultaneously in the brain, allowing for flexible knowledge at multiple levels of specificity.

**Decision letter after peer review:**

Thank you for submitting your article "Model-based fMRI reveals co-existing specific and generalized concept representations" for consideration by *eLife*. Your article has been reviewed by three peer reviewers, including Morgan Barense as the Reviewing Editor and Reviewer #1, and the evaluation has been overseen by Timothy Behrens as the Senior Editor. The following individual involved in review of your submission has agreed to reveal their identity: Alexa Tompary (Reviewer #2).

The reviewers have discussed the reviews with one another and the Reviewing Editor has drafted this decision to help you prepare a revised submission.

Summary:

Bowman et al. investigate a long-standing debate in cognitive science: whether categories are represented by exemplar category members or by prototypes that are extracted from individual members. In this experiment, participants undergo fMRI as they learn to categorize stimuli that vary along 8 dimensions. Model fitting is used to characterize participants' decisions and neural responses as supported by either the prototype or exemplar model of categorization. They find that there are separate sets of regions that relate with the prototype-based and exemplar-based models, suggesting that the two can coexist as strategies employed by different systems.

The reviewers were very enthusiastic about the question and the model-based fMRI approach, and we all appreciated the attempt to reconcile disparate findings in the field. That said, there were serious concerns about some of the analyses, consistency of results, and thus the ultimate implications for the bigger picture. We thought that the paper would benefit from a more extensive comparison to the authors' past work, as well as a deeper synthesis of the broader debate in general. The methods could be better grounded in the theoretical issues at stake. The reviewers were initially split on how to proceed. However, the reviewers were unanimous in seeing great value in the theoretical question at hand, and in the opinion that model-based fMRI has a lot of potential for our field in general. Thus, we wanted to give the opportunity for the authors to revise the manuscript, with the clear advice that any revision must address these concerns if it is to be successful. Below is a synthesis of the reviewers' comments and the discussion that emerged from our consultation session.

Essential revisions:

1) There were concerns that the pattern similarity analysis had underlying flaws. Specifically, the two comparisons (of item representations and category representations) do not appear to be on equal grounding. An item will have a highly similar representation to itself because it is visually identical to itself. This reviewer felt that it was not surprising (or necessarily interesting) that there are regions that show higher item self-similarity than cross-similarity, and we would hesitate to say these regions contain a more abstracted representation of the item or exemplar versus just responding to identical visual or perceptual processing of a given image. The category representation task is much harder (showing differences based on perceptually matched but learned category distinctions), and indeed it doesn't appear to come out from these data. What might be a stronger test than looking at pattern similarities during this learning period, would be to look during the test periods. The authors mention concerns about motor responses, but those could be regressed out. It was suggested that the authors could create two representational dissimilarity matrices based on the two models (exemplar and prototype) for each stimulus. For the exemplar model, an ROI pattern would be predicted with similarity to exemplars previously shown, while for the prototype model, an ROI pattern would be predicted with similarity to the average. This should be a more direct test.

2) Although the pattern similarity analyses were not the primary analyses in the paper, there were two very serious concerns about the primary model-based fMRI analyses. These will be essential to address if a revision is to be successful:

a) How correlated are the prototype model predictions and the exemplar model predictions? It seems like they might be highly correlated (e.g., very distant items would be distant from the prototype as well as previous exemplars). If so, then the authors may need to correct for multicollinearity in the GLMs – their separate contributions cannot be interpreted if they are correlated (because their shared variance is an arbitrary split between the two).

b) The within-ROI analyses were all one-sample t-tests only for the hypothesized model in a hypothesized region of interest. However, it is equally important to know: 1) is there evidence for the other model in that region as well (e.g., is there both prototype and exemplar information in the VMPFC, and if so, how can one interpret their co-existence?) and 2) is one model stronger than the other in a region (e.g., if the VMPFC doesn't have significantly higher prototype model fit than an exemplar one, can one really say it is a prototype-specific region?)

3) We discussed the fact that the paper presents the evidence for exemplar-based processing is not strong. Behaviorally there is significantly more evidence for the prototype model, and although it appears in the hypothesized regions during the 2nd phase, it doesn't appear in the 1st phase or the final phase. The pattern similarity results also do not directly address exemplar-based coding, for the reasons mentioned above. The above-mentioned RSA may help with this, but there was concern that the paper presents only weak evidence for the existence of an exemplar representation. On the flip side, we were concerned that the prototype model does show significantly higher fits than the exemplar model in most regions when pitted directly against each other (vs. the current analysis which just compares the hypothesized model to 0 – per Comment 2b above). As such, there was concern that the final set of results do not show a message that is consistent across timepoints, behaviour, and analyses. Pending the results of additional analyses, we felt that the conclusions should be tempered, or it may simply be the case that the results are not strong enough to warrant publication in *eLife*.

4) In the authors' prior paper (J Neuro 2018), participants were classified according to whether their responses were well fit by the exemplar model, prototype model, or neither. Here, these fits are displayed on a continuous scale rather than classifying participants according to their dominant strategy, and it seems that there is wide variability in these fits and with a majority of subjects either well or poorly fit by both (participants along the diagonal line).

The reviewers were curious how participants would be classified according to their prior approach, as a potential means to answer the question of whether performance in this dataset is indeed more exemplar-based than in the author's prior dataset. We agreed that the manuscript would benefit from a justification of why the authors switched to this analysis rather than the classification approach they had previously adopted. If possible, it would be nice to show both for a more direct comparison.

5) It was unclear how a participant could be a good fit to both models, as they should predict opposing behaviors (e.g. an advantage for old items over new ones in the exemplar model versus no difference in the prototype model, when testing items 2 features away from their prototype). An example or a high-level description of what a good fit to both models may look like would be helpful here.

6) Related to the first point, the authors state that task demands and/or category structures could bias categorization to be either exemplar or prototype based. To get at both, the authors used a category structure that is supported mostly by a prototype model, but use a task that is meant to promote exemplar encoding. However, on average participants' decisions were still better fit by the prototype model (although again, most people fall along the diagonal). And from Figure 4, there seem to be very few participants that were better fit by the exemplar model.

With this in mind, is there any evidence that the observational task actually shifted participants to be more exemplar-based? If not, why is there evidence of exemplar representations in LO/IFG/Lateral parietal in this experiment and not in the authors' earlier paper, where ~10% of participants were better fit by an exemplar model? Are there any other differences between the two procedures or stimulus sets that could account for this? Regardless of the differences between the experiments, the notion that exemplar-based representations exist in the brain even though they are not relevant for behavior is worth engaging with in more detail in the Discussion.

7) In general, it was felt that the paper could use more depth. Many of the citations regarding the prototype vs. exemplar debate are very old (~30+ years old) and the more recent papers mentioned later focus more on modeling and neuroimaging. Could the authors describe work that consolidates this debate? Also, we felt that there could be more elaboration on the Materials and methods in the main manuscript, specifically as they relate to the theoretical questions at stake. For example, what are the general principles underlying these models and what different hypotheses do they make for this specific stimulus space?

8) In addition to engaging more with the literature, we wished for more of a discussion of the mechanistic reasons behind why these ROIs have separate sensitivity to these two models, as well as what the present results show about what underlying computations are occurring in the brain. To quote directly from our consultation session: "I think some attempt at a mechanistic explanation may also be necessary. (Though I understand this can be delicate because they may want to avoid reverse inference.) One thing that stands out to me here is that LO is claimed to be a region showing exemplar representations. LO is believed by many (most?) to be a "lower level" visual area (e.g., in comparison to PPA which may have more abstract information) that represents visual shape information. So even if LO shows an exemplar representation, this could be because the item is more similar in shape to other items that were seen (the exemplars) versus a prototype that was not seen. So, a purely low-level visual account could possibly explain these results, rather than something deeper about mental representations. Thus I am also concerned about whether these findings necessarily mean something deep reconciling this larger theoretical debate, or may reflect some comparison of low level visual features across exemplars." In short, we all agreed that a deeper discussion of the neural computations and mechanisms would improve the contribution of the current paper considerably.

9) Six participants were excluded for high correlated prototype and exemplar regressors – can the authors provide a short of explanation of what pattern of behavioral responses would give rise to this? And what is the average correlation between these regressors in included participants?

10) For behavior, the authors investigated performance as it varied with distance from prototypes. It would also be interesting to investigate how behavior varies as distance from studied exemplars.

11) Did the authors find different weighting on different manipulated features?

12) The manuscript states that for the exemplar model, "test items are classified into the category with the highest summed similarity across category exemplars." One reviewer wondered whether this is a metric that is agreed upon across the field, as they would have anticipated other (possibly non-linear) calculations. For example, could it be quantified as the maximum similarity across exemplars?

13) Is there any way to explore whether the model fits from the neural data can be used to explain variance in participants' behavioral fits? For instance, do neural signals in vmPFC and anterior hippocampus better relate to behavior relative to signals in LO, IFG and lateral parietal for participants with better prototype fits? There may not be sufficient power for this so this was suggested only as a potential exploratory analysis.

[Editors' note: further revisions were suggested prior to acceptance, as described below.]

Thank you for resubmitting your article "Tracking prototype and exemplar representations in the brain across learning" for consideration by *eLife*. Your revised article has been reviewed by two peer reviewers, and the evaluation has been overseen by a Reviewing Editor and Timothy Behrens as the Senior Editor. The following individual involved in review of your submission has agreed to reveal their identity: Alexa Tompary (Reviewer #2).

The reviewers have discussed the reviews with one another and the Reviewing Editor has drafted this decision to help you prepare a revised submission.

We all agreed that the revised paper was substantially improved. In particular, the removal of the pattern similarity analyses made the paper sharper and more straightforward, as well as eliminated the methodological concerns that were highlighted in our original reviews. The pairwise t-tests of prototype vs. exemplar fits in the ROIs has clarified the results, while still providing good support of the authors' conclusions. The relationship between prototype and exemplar models is much clearer and is thoughtfully discussed, and the added text in the Introduction and Discussion have nicely framed the paper and motivated the analyses. In general, the research reads as being much more impactful and the current work much stronger.

Although the paper is much improved, one concern still remains regarding the correlation between exemplar and prototype regressors. To quote reviewer #2 specifically:

"I'm convinced that mathematically, the regressors are not highly correlated and the explanation in the authors' response regarding the most distant items is helpful in getting an intuition of how this can be. At the same time, I'm still trying to understand how the correlation between regressors is on average only r=.3 when the behavioral fits themselves are so highly correlated across participants. If there is a way to clarify this, I think that would go a long way in helping readers get on board with the idea that both exemplar and prototype representations can exist in the brain.

Incorporating some of the authors' explanation from the response letter could be useful for this. In particular, it was nice that they explicitly laid out how the advantage for old items over new ones would result in a better fit to an exemplar model but still a decent fit to an exemplar model because accuracy for old/new items at distance 2 fall between accuracy for items with distances of 1 an 3. Their response to Comment 7 seems like it could be informative to include too – although here they talk about the models' differences in terms of confidence, which isn't explained in the manuscript."

---

## [Author Response]

Essential revisions:1) There were concerns that the pattern similarity analysis had underlying flaws. Specifically, the two comparisons (of item representations and category representations) do not appear to be on equal grounding. An item will have a highly similar representation to itself because it is visually identical to itself. This reviewer felt that it was not surprising (or necessarily interesting) that there are regions that show higher item self-similarity than cross-similarity, and we would hesitate to say these regions contain a more abstracted representation of the item or exemplar versus just responding to identical visual or perceptual processing of a given image. The category representation task is much harder (showing differences based on perceptually matched but learned category distinctions), and indeed it doesn't appear to come out from these data.

We thank the reviewers for noting that the rationale and interpretation for the pattern similarity analyses were not clear in our original manuscript. We were debating whether or not to include the pattern similarity analysis in our original submission as the additional experimental phases (training runs, interim tests) already made the methods and results quite long compared to our previous 2018 paper. Additionally, because the model-based fMRI and the pattern similarity analyses focus on very distinct notions of what “abstract” category representations can mean, we were worried that it may lead to confusion. During the revisions, we decided to drop the pattern similarity analyses from the paper to streamline it and allow us to better flesh out the model-based analyses. Nevertheless, we do think that the pattern similarity analyses could be interesting in their own merit and hope to explain it better in our response than we did in the original submission.

Regarding category representations, we agree that the bar we set for the category representation operationalization was high. This was intentional: we were specifically interested if we can find category representations under the “strict” conditions of equated physical similarity within- and across- categories. We realize that a positive outcome would have been much more interesting from the reader’s perspective, but we reported the negative outcome nevertheless.

For the reviewers’ information, the lack of category representations is interesting to us. The reason is that we have another study currently in progress that uses different stimuli (face blends) in which we *do* find such category representations in a strict sense. Moreover, they emerge over the course of learning across many brain regions, including visual cortices. We felt that the comparison between the studies may help interpret both findings. For example, category learning often includes learning of what features are relevant vs. irrelevant for category membership. Greater within-category and smaller between-category similarity may then result from the hypothesized stretching and shrinking of perceptual space along those relevant and irrelevant perceptual dimensions, as one learns to attend to relevant information and ignore irrelevant information. As all stimulus features are relevant in the current study, one cannot achieve increase of perceived within-category similarity or decrease of perceived between-category similarity by ignoring irrelevant stimulus dimensions. This may explain why category representations in the strict sense were not found in the current study (although they could still emerge in higher cognitive regions). However, we cannot reference the *in-progress* study in the current manuscript, leaving the category representations to remain a seemingly non-informative null finding at this time.

Regarding item representations, we did not aim to argue that the observed item representations are abstract and apologize if it came out that way. However, we would also like to note that item representations are not necessarily trivial, nor they seem purely visually driven in our data. First, there were no stable item representations early in learning and instead they only emerged late in learning. This indicates that the formation of stable representations of individual items across repetitions involves learning and memory [as also illustrated in e.g., Xue et al., (2010, Science)]. Furthermore, item representations also weren’t strongest in a perceptual ROI (LO) but instead were strongest in the lateral prefrontal and especially lateral parietal cortices. We thought this was interesting in light of research on memory fidelity and protection from interference among similar memories, which motivated these regions of interest in the first place (we expand on this point in response to comment 11 as well). However, we realize we did not verbalize this well in the manuscript.

During the revision, we were again torn whether to keep the pattern similarity analyses and expand on their rationale and interpretation, or whether to drop them. We ended up dropping them for brevity and other reasons listed above, and agree that the manuscript is indeed much easier to digest without these secondary analyses. This also gave us more space to expand on the model-based results in response to other comments, as we outline below.

What might be a stronger test than looking at pattern similarities during this learning period, would be to look during the test periods. The authors mention concerns about motor responses, but those could be regressed out.

We agree that it would be interesting to do the pattern similarity analysis during the tests but opted against it initially because we wanted to focus on the strictest form of category representation – increased within-category representational similarity despite equal physical similarity. One challenge of using both model-based MRI and pattern similarity analyses is that it is difficult to find a category structure and a set of category examples that are well suited to both approaches. The stimulus set that we selected for the test periods was optimized for model-based fMRI and included a variety of examples that differed in their distance from category prototypes and from the training examples. This variability helped generate prototype and exemplar regressors that were not too correlated with one another and included a range of model-predicted values for the parametric modulation to track. However, this stimulus set was less well suited to pattern similarity analyses because the physical similarity between and across categories is no longer equated (as it was equated among the training stimuli). Thus, there are fewer PSA comparisons where physical similarity was matched within and across categories for the “strict” category representation analysis.

However, for the reviewers’ information, we ran category PSA for the test portions and include the results in Author response image 1. First, we focused on testing the “strict” category representations analogous to the original learning analysis, using only comparisons of items 4 features apart to equate physical similarity within- and between categories (left panel). No region showed reliable category representations during the interim or final tests. Second, we ran a “normal” category PSA and included all pairwise comparisons of all test stimuli (ignoring that two stimuli from the same category would tend to have also higher physical similarity). Here, we see that in VMPFC and LO, items within a category tend to be represented more similarly than items from different categories by the final test.

Because we no longer report any pattern similarity analyses, we did not include these results in the revised manuscript, but wanted to share them with the reviewers.

It was suggested that the authors could create two representational dissimilarity matrices based on the two models (exemplar and prototype) for each stimulus. For the exemplar model, an ROI pattern would be predicted with similarity to exemplars previously shown, while for the prototype model, an ROI pattern would be predicted with similarity to the average. This should be a more direct test.

We thank the reviewers for engaging deeply with our paper and coming up with additional suggestions for analyses. It would indeed be interesting to develop an analysis that directly compares neural RDMs with model-derived RDMs, in the tradition of other RSA work (e.g., Kriegeskorte, 2008, Frontiers).

The first approach we tried in response to this comment was to construct the model-based RDMs by computing predicted perceived distance between all pairs of stimuli using the attention weights derived from the respective models. This was the approach taken by Mack et al., 2013. Let’s consider a hypothetical example using 4-dimensional stimuli, prototype-based weight estimates being [.5.5 0 0] and exemplar-based weight estimates being [.25.25.25.25]. Then the dissimilarity between two stimuli, S1=[1 1 1 1] and S2=[1 1 0 0], would be zero according to the prototype model (because it assumes that features 3 and 4 are ignored) but it would be 0.5 per the exemplar model (because it assumes that features 3 and 4 are attended to just as much as features 1 and 2). Model-based RDMs can be constructed this way for all pairs of stimuli and compared with neural-based RDMs, as done by Mack et al.

Unfortunately, when we tried this analysis, we found it unsuitable for differentiating between the models in our data. Specifically, the two models produce highly correlated feature weight estimates (mean r = 0.85, range 0.53-1.00, see the histogram of correlation values in Author response image 2). We have previously shown in a separate large behavioral sample that feature weight estimates are well-correlated between the models when using the current stimuli and a range of category structures (Bowman and Zeithamova, 2020), so the model agreement on feature weights is not unique to this particular data set. When the prototype model fits indicate that Subject X ignored dimensions 3 and 4 and paid most attention to dimension 2, then the exemplar model typically indicates the same. From one perspective, this is a good validation that model fitting does what it should and the attention weight estimates are reliable. However, it also means that the prototype model-predicted and exemplar model-predicted RDMs are highly similar in our data and cannot serve to resolve which model better matches the neural RDMs. This was not the case in the Mack paper where the attention weight estimates differed enough between models to generate distinct predictions. We speculate that because the training stimuli in Mack et al. were less coherent and not centered around a prototype, the attention weight estimates per the prototype model were perhaps noisier. As we have a larger number of test stimuli in our studies and both exemplar and prototype strategy are in principle suitable for the current stimulus structure, both models may have sufficient information to estimate the participant’s attention to different features well-enough. Because this analysis is essentially a physical similarity analysis with weighing of features added, the model-based similarity predictions don’t differ between the prototype and exemplar model when the feature weight estimates are the same.

**Author response image 2. respfig2:** 

The wording of the comment also suggests an alternative analysis where the predicted similarity between two stimuli could be computed through a comparison of each stimulus to the exemplars/prototypes. We were considering such an approach, but it may not be possible to generate model-based similarity predictions this way. For instance, let’s consider what a similarity prediction should be for a pair of stimuli, S1 and S2, based on their similarity to the prototypes. Let’s assume that both S1 and S2 are 2 features from the prototype A and 6 features from prototype B. Even though S1 and S2 are both 2 features away from Prototype A, they can still differ by zero, 1, 2, 3 or 4 features from each other. Thus, it is unclear what prediction we should make about how similar S1 and S2 are to each other based on how similar they are to the two prototypes. The same holds for the exemplar model, but gets even more complicated given that a stimulus may be highly similar to an existing exemplar representation through similarity to very different category exemplars. Because this ambiguity in what model-based similarity predictions should be, we did not aim to implement this particular analysis.

2) Although the pattern similarity analyses were not the primary analyses in the paper, there were two very serious concerns about the primary model-based fMRI analyses. These will be essential to address if a revision is to be successful:a) How correlated are the prototype model predictions and the exemplar model predictions? It seems like they might be highly correlated (e.g., very distant items would be distant from the prototype as well as previous exemplars). If so, then the authors may need to correct for multicollinearity in the GLMs – their separate contributions cannot be interpreted if they are correlated (because their shared variance is an arbitrary split between the two).

We appreciate this concern and took it into consideration when designing the task and during the analysis. It is indeed the case that stimuli further from old exemplars will also tend to be further from the prototypes. That’s also why the behavioral predictions of the two models are generally correlated and one can use either representation to solve the task. But this correlation will not be perfect, allowing distinct-enough predictions from the two models to differentiate them in behavior and in the brain in most cases.

Let’s take the example from the comment, focusing on the most distant items. The furthest a new exemplar in our task can be from the prototype is 3 features (4 features would make it equidistant from the two prototypes, and such stimuli were not used). When designing the task, we aimed to use a stimulus set that would be unlikely to generate highly correlated regressors and made sure that stimuli furthest from the prototype are *not always* far away from all the training exemplars. We used 8 new final test stimuli at distance 3 from the prototype A. Out of the 8 stimuli, there will be some that are near an old exemplar and some that are not near any of the old exemplars. The prototype model will have the same prediction for all distance 3 stimuli but the exemplar model will not. This will be also the case for stimuli at all other distances from prototype A. Because we varied the test stimuli to include all distances from the prototypes, and because within each distance to the prototype there is variability in how far the stimuli are from the old exemplars, the structure is set up to facilitate dissociation between the model predictions. And, in most cases, this variability among stimuli indeed resulted in sufficiently distinct predictions from the two models to permit analysis.

Although the objective structure was set up to facilitate dissociability, the reviewers are correct that correlated predictors are still a potential issue. We have control over the physical distances among the stimuli. However, each participant’s attention to different stimulus dimensions affected perceived similarity, and we do not have a control over that. For example, if physical distance is 2, but participant’s responses indicate that they didn’t really pay attention to those two features that happen to be different, the perceived distance between those stimuli becomes zero. Thus, the attentional parameters that we estimated for each subject affected the actual correlation between model predictions.

For that reason, we always check for correlation of regressors for each subject after the individual model fitting and exclude participants for whom the correlation is too high to permit the GLM analysis. In this study, it was 5 participants, as originally reported in the Participants section and as noted in Comment 12. In Author response image 3, we plot the histogram of the correlation values across all runs and all subjects that were included in the analysis.

**Author response image 3. respfig3:** 

The majority of correlations were small or moderate. However, there were some runs with the absolute correlation being relatively high. While none of these runs were flagged by FSL as rank-deficient or too highly correlated to estimate, we wanted to verify that our results were not affected by the correlation between the regressors. For example, it could be that the lack of exemplar correlates in the final test was due to the shared variance being mis-assigned. Thus, we re-created in Author response image 4 the final test ROI x Model analysis in the subset of subjects (n = 20) whose regressor correlations fell between +/- 0.5 for all final test runs (with most runs much closer to zero than.5). The overall pattern of results remained the same: VMPFC and anterior hippocampus tracked prototypes, and there were no above-chance exemplar correlates. Thus, it does not appear that the specific pattern of results was due to the limitations of running GLM (multiple regression) with correlated predictors.

**Author response image 4. respfig4:** 

We did not re-compute the ROI x model analysis for the second half of learning because all runs in all subjects during this phase had regressor correlations between (-.32, +.46), with the vast majority close to zero. Thus, the presence of both prototype and exemplar correlates across regions during this phase does not seem to be an artefact of correlated regressors.In the revised manuscript, we expanded the model description and regressor generation section to better explain how the potential correlation between regressors was taken into account:

“The modulation values for each model were computed as the summed similarity across category A and category B (denominator of Equation 3) generated under the assumptions of each model (from Equations 1 and 2). […] We verified that the pattern of results remained the same when analyses are limited to participants with absolute correlations below.5 in all runs, with most correlations being quite small.”

b) The within-ROI analyses were all one-sample t-tests only for the hypothesized model in a hypothesized region of interest. However, it is equally important to know: 1) is there evidence for the other model in that region as well (e.g., is there both prototype and exemplar information in the VMPFC, and if so, how can one interpret their co-existence?) and 2) is one model stronger than the other in a region (e.g., if the VMPFC doesn't have significantly higher prototype model fit than an exemplar one, can one really say it is a prototype-specific region?)

The reported analyses were based on our primary goal to replicate prototype correlates in VMPFC and AHIP from our 2018 paper and test whether we can replicate exemplar correlates in IFG, lateral parietal, and LO regions reported by Mack et., 2013. However, we agree that the analyses more directly testing a potential regional dissociation would be informative and should be reported as well. We were going back and forth about these analyses during the initial submission and ended up removing them to streamline the Results section. We have added these analyses to the Results section and briefly summarize them here:

1) We did not observe a region with significant prototype AND significant exemplar signals. As a side note, we do think that such a result is theoretically possible and offer a speculation about it at the end of the response to this comment, in case the reviewers are interested.

2) The direct comparison of prototype vs. exemplar signal did not reach significance in the predicted prototype regions during the interim tests (anterior hippocampus p=.19; VMPFC p=.11) but was significant during the final test in the anterior hippocampus (p=.015) and marginal in VMPFC (p=.053). The exemplar signal was significantly greater than the prototype signal in the majority of predicted exemplar regions (lateral parietal, IFG, PHIP; all p <.04) and marginal in LO (p=.08) during the interim tests. During the final test, there was no reliable exemplar signal and we also did not find any exemplar > prototype region (all p’s>.5). These results are now reported in detail in the revised manuscript.

In line with these results, we have revised our conclusion regarding the VMPFC. First, we emphasize that it contains generalized category representations abstracted across exemplars, akin to its role in memory integration across events. Second, we note that whether VMPFC is prototype-specific is still an open question in our view, and we hope to dive into that question in future research. The most relevant revised discussion text can be found here:

“Prior work has shown that the hippocampus and VMPFC support integration across related experiences in episodic inference tasks (for reviews, see Schlichting and Preston, 2017; Zeithamova and Bowman, 2020). […] Thus, it remains an open question whether representations in VMPFC are prototype specific or instead may reflect some mix of coding.”

As a side note, we think that the VMPFC in particular could in principle show both types of signal (prototype and exemplar) and that such a result could be meaningful. For example, exemplar-based category representations could emerge in VMPFC, perhaps with a less coherent category structure. We have behaviorally shown that less coherent training exemplars are less likely to produce a prototype representations (Bowman and Zeithamova, 2020). But a category label could serve as overlapping information linking category exemplars even if exemplars are too distinctive to produce a coherent prototype. This would resemble arbitrary overlapping pairs as used in associative inference tasks: S1-Badoon, S2-Badoon. If we had found exemplar signals in VMPFC, we would be curious to follow up to find out whether distinct representations form in distinct subregions of VMPFC (perhaps akin to separated (AB,BC) vs. integrated (ABC) representations identified in Schlichting et al., 2015). Or perhaps the same neural region would sometimes form exemplar and sometimes prototype representations, depending on the task or on the participant. Given that we did not actually find a region that would show both signals, we did not reproduce the above discussion in the paper itself but are including it here to answer the reviewers’ question.

3) We discussed the fact that the paper presents the evidence for exemplar-based processing is not strong. Behaviorally there is significantly more evidence for the prototype model, and although it appears in the hypothesized regions during the 2nd phase, it doesn't appear in the 1st phase or the final phase. The pattern similarity results also do not directly address exemplar-based coding, for the reasons mentioned above. The above-mentioned RSA may help with this, but there was concern that the paper presents only weak evidence for the existence of an exemplar representation. On the flip side, we were concerned that the prototype model does show significantly higher fits than the exemplar model in most regions when pitted directly against each other (vs. the current analysis which just compares the hypothesized model to 0 – per Comment 2b above). As such, there was concern that the final set of results do not show a message that is consistent across timepoints, behaviour, and analyses. Pending the results of additional analyses, we felt that the conclusions should be tempered, or it may simply be the case that the results are not strong enough to warrant publication in eLife.

We agree with the reviewers that the original submission may have sold the idea of co-existing representations too strongly. As noted to the Comment 4b, one piece of supporting evidence was added: an analysis that shows that the exemplar correlates in the hypothesized exemplar regions were not only above chance, but also reliably greater than prototype correlates. While this helps increase confidence in the exemplar findings during 2^nd^ phase, it does not alleviate the concern that the exemplar correlates are only apparent in part of the task and do not carry consistently across timepoints. To better reflect the limits of exemplar evidence, we have revised the title of the manuscript to “Tracking prototype and exemplar representations in the brain across learning”. We have made revisions to the Abstract and throughout the discussion to give it the nuance warranted by the results. Removing the RSA analyses has also given us space to expand our discussion of the model-based results.

4) In the authors' prior paper (J Neuro 2018), participants were classified according to whether their responses were well fit by the exemplar model, prototype model, or neither. Here, these fits are displayed on a continuous scale rather than classifying participants according to their dominant strategy, and it seems that there is wide variability in these fits and with a majority of subjects either well or poorly fit by both (participants along the diagonal line).The reviewers were curious how participants would be classified according to their prior approach, as a potential means to answer the question of whether performance in this dataset is indeed more exemplar-based than in the author's prior dataset. We agreed that the manuscript would benefit from a justification of why the authors switched to this analysis rather than the classification approach they had previously adopted. If possible, it would be nice to show both for a more direct comparison.

We thank the reviewers for this suggestion. We agree that the classification approach provides a nice way to visualize the behavioral model fits and offers the most straightforward way to evaluate whether we succeeded in shifting participants’ strategy compared to our prior paper. We are now including it in the manuscript. For the reviewers’ information, we used the alternative report because we were asked to report model fits using a scatter plot for our recent behavioral paper (Bowman and Zeithamova, 2020) and it also matched how Mack et al., 2013, visualized their model fits (we also note why the scatter plots are a popular way to display model fits in the response to Comment 7). We then ended up not including the model classification approach in the initial submission to shorten the paper but are happy to include them in the revision.

From the classification analysis above, we see that we succeeded in shifting participant’s strategy only partially. In our prior study, which had shorter, feedback-based training and only included a final test, 73% participants were best fit by the prototype model. In the current study, prototype model wins by a smaller margin and there are more participants who are comparably fit by both models. However, the prototype model still dominates behavior starting with second part of training.

In the revised manuscript, we added the strategy classification analysis and Figure 4. The relevant Materials and methods and Results are reprinted below:

Materials and methods:

“We also tested whether individual subjects were reliably better fit by one model or the other using a permutation analysis. […] We then compared the observed difference in model fits to the null distribution of model fit differences and determined whether the observed difference appeared with a frequency of less than 5% (α=.05, two-tailed). Using this procedure, we labeled each subject as having used a prototype strategy, exemplar strategy, or having fits that did not differ reliably from one another (“similar” model fits) for each phase of the experiment.”

Results:

“Figure 4D-F presents the percentage of subjects that were classified as having used a prototype strategy, exemplar strategy, or having model fits that were not reliably different from one another (“similar”). In the first half of learning, the majority of subjects (66%) had similar prototype and exemplar model fits. In the second half of learning and the final test, the majority of subjects (56% and 66%, respectively) were best fit by the prototype model.”

The Results section now also includes additional explanation of the alternative, continuous measure visualization using the scatter plots, which are detailed in response to Comment 7.

Finally, we added a discussion of the modest strategy shifts between the current and 2018 studies, as we detail in response to Comment 8.

Please note that in our original 2018 paper, we used the label “neither” in the sense that neither model outperformed the other model. However, we found out that it was confusing as it sounded like neither model fit the behavior well. That was not the case – both models outperformed chance (in our 2018 paper as well as here). We are now using “similar fit” instead of “neither” to label the participants whose exemplar and prototype model fits are comparable in order to avoid confusion about this point.

5) It was unclear how a participant could be a good fit to both models, as they should predict opposing behaviors (e.g. an advantage for old items over new ones in the exemplar model versus no difference in the prototype model, when testing items 2 features away from their prototype). An example or a high-level description of what a good fit to both models may look like would be helpful here.

We thank the reviewers for noting that in our attempt to explain dissociable predictions from the two models, we failed to note their similarities. In general, the two models will predict similar category labels, albeit with different confidence. For instance, prototype A will be predicted with high confidence to be a category A item by the prototype model. But it will be also predicted with moderate confidence to be a category A item by the exemplar model, as it is also closer to category A training exemplars than category B training exemplars. Category A training exemplars will be classified with high confidence as category A items by the exemplar model, but also classified with moderate confidence as category A items by the prototype model because they are only 2 features away from prototype A but 6 features away from prototype B. When participants are just guessing or pressing random buttons, neither model will do well. But as participants learn and do better on the task, the fit of both models will tend to improve. Even if a participant had a purely prototype representation and a perfect prototype fit, the exemplar model would still predict behavioral responses (based on the similarity to category exemplars) reasonably well and definitely significantly better than chance. Conversely, if a participant had a purely exemplar representation, items close to the old exemplars of category A will be – on average – also closer to the prototype A than prototype B. Thus, the prototype model will still fit quite well, but not as well as the exemplar model.

Because the model fits are usually correlated, they are often plotted as a scatter plot (as we did in the original Figure 4 and explained in response to Comment 6), with the diagonal line representing equal fits. Participants above the line are better fit by one model, participants below the line are better fit by the other model, and participants on the line are fit equally well by both models. (Our Monte Carlo procedure allows us to make a little more principled decision with respect to when to call the fits “equal”). Participants with higher (mis)fit values are those who learned less and are guessing on more trials, which will mean that both models will fit relatively poorly. And as noted above, the more consistent the responses become with an underlying representation (prototype or exemplar), the better fit for both models will tend to be. This is the reason why the exact fit value for one model is not sufficient to determine the participant’s strategy; only the relative fit of one model compared to the other model is diagnostic. In some cases, the responses will be highly consistent with both models and we end up unable to determine the strategy.

Up to this point, we discussed the mathematical reasons for why a great fit of one model will be likely accompanied by a decent fit of the other model. However, it is worth noting that there may also be cognitive reasons why a participant’s behavior may be well fit by both models. We don’t see prototype vs. exemplar representations to be necessarily an either-or scenario; it would be plausible that a participant forms a generalized representation (prototype) that guides categorization decisions but also has memory for specific old instances that can also inform categorization. For example, consider a participant whose behavior mirrors what we see across the group: they have an accuracy gradient based on the distance from the category prototypes but also have an accuracy advantage for old items compared to new items at distance 2 (Figure 3B in the current manuscript, also observed in 2018 paper). The reviewers are correct that the prototype model does not predict an old/new difference. However, accuracy for both old and new distance 2 items still falls between distance 1 and distance 3 items, consistent with the prototype model predictions. Thus, the observed old/new advantage will not cause a serious reduction in model fit for the prototype model as both old and new items at a given distance fall along the expected distance gradient. At the same time, the old > new advantage suggests that the prototype model does not offer the whole story.

In the revised manuscript, we included more information about the models to clarify why behavioral model fits in general track one another and why both models may fit behavior well:

“Figure 4A-C presents model fits in terms of raw negative log likelihood for each phase (lower numbers mean lower model fit error and thus better fit). Fits from the two models tend to be correlated. If a subject randomly guesses on the majority trials (such as early in learning), neither model will fit the subject’s responses well and the subject will have higher (mis)fit values for both models. […] In such cases, a subject may be relying on a single representation but we cannot discern which, or the subject may rely to some extent on both types of representations.”

We have also provided more motivation for the idea that prototype and exemplar representations may co-exist in the Introduction:

“It is possible that the seemingly conflicting findings regarding the nature of category representations arose because individuals are capable of forming either type of representation. […]Thus, under some circumstances, both prototype and exemplar representations may be apparent within the same task.”

6) Related to the first point, the authors state that task demands and/or category structures could bias categorization to be either exemplar or prototype based. To get at both, the authors used a category structure that is supported mostly by a prototype model, but use a task that is meant to promote exemplar encoding. However, on average participants' decisions were still better fit by the prototype model (although again, most people fall along the diagonal). And from Figure 4, there seem to be very few participants that were better fit by the exemplar model.With this in mind, is there any evidence that the observational task actually shifted participants to be more exemplar-based? If not, why is there evidence of exemplar representations in LO/IFG/Lateral parietal in this experiment and not in the authors' earlier paper, where ~10% of participants were better fit by an exemplar model? Are there any other differences between the two procedures or stimulus sets that could account for this? Regardless of the differences between the experiments, the notion that exemplar-based representations exist in the brain even though they are not relevant for behavior is worth engaging with in more detail in the Discussion.

The reviewers’ point that the difference between the behavioral and neural model fits should be discussed in more detail is well taken. First, as the reviewers noted (and as we discussed in Comment 6) the strategy shift turned out to be modest. Despite that, we found exemplar correlates (at least in the interim tests after the second half of training) in the present study while we did not find them in our 2018 study. Why is that? First, we need to note that although our 2018 study did not find any above-threshold exemplar regions when using a standard corrected threshold, we found some parts of the brain, including lateral occipital and lateral parietal region, that tracked exemplar predictions at a lenient threshold (z=2, FDR cluster correction p=.1). We conducted this exploratory lenient threshold analysis in an attempt to reconcile our findings with that of Mack et al., 2013, and reported it on p. 2611 (last Results paragraph) of our 2018 J Neuro paper. Thus, the present results are not entirely different from our prior study in that there was some evidence of exemplar-tracking despite better fit of the prototype model to behavior.

One possible reason for observing exemplar correlates in the current study may be that the observational learning was actually successful in promoting the exemplar representation, even though the prototype strategy still dominated behavior. There are fewer of the pure “prototypists” and more subjects that are comparably fit by both models, providing some indication that there was a small strategy shift. The small shift in strategy may have been sufficient for the previously subthreshold exemplar correlates to become more pronounced.

The structure of the training was indeed the primary difference between the two studies. Our prior study included a single feedback-based training session outside the scanner. The present study included cycles of observational study runs (to promote exemplar memory) that were followed by interim generalization tests (to measure learning and estimate model fits). We also used a different set of cartoon animals, but their feature space is very similar and the structure of the training items was the same across studies. While the training portion was necessarily longer in the current study because it was scanned and included both observational learning and interim tests, the final tests were nearly identical across studies. Thus, the differences in training are likely the most important from the learning perspective.

Importantly, we agree with the reviewers that the prototype dominance in categorization behavior may not preclude the existence of exemplar-specific memory representations. Exemplar-based representations may form in the brain even though they are not immediately relevant for the task at hand. Exploring the possibility that specific and generalized memories form in parallel is of high interest to our lab, and we have behaviorally studied both specific memories after generalization learning (Bowman and Zeithamova, 2020) and generalization after learning focused on specific details (Ashby, Bowman and Zeithamova, 2020, Psychology Bulletin and Review). Thus, we are eager to dedicate more discussion to this topic.

In the revised manuscript, we expanded the comparison between the current study and our 2018 study throughout the discussion. We also discuss the existence of neural exemplar-based representations in the context of prototype dominance in behavior:

“In designing the present study, we aimed to increase exemplar strategy use as compared to our prior study in which the prototype model fit reliably better than the exemplar model in 73% of the sample (Bowman and Zeithamova, 2018). […] The present results show that these parallel representations may also be present during category learning.”

7) In general, it was felt that the paper could use more depth. Many of the citations regarding the prototype vs. exemplar debate are very old (~30+ years old) and the more recent papers mentioned later focus more on modeling and neuroimaging. Could the authors describe work that consolidates this debate?

We agree that our paper could benefit from broader connections to the existing literature and deepening of the discussion. We have revised the Introduction and especially the Discussion and incorporated a number of additional references relevant to the current work (noted in blue in the manuscript). We specifically aimed to include more recent papers focusing on prototype and/or exemplar models (e.g., Dubè, 2019; Thibaut, Gelaes and Murphy, 2018; Lech, Güntürkün and Suchan, 2016). Unfortunately, to our knowledge, they are not very prevalent and the best attempts at consolidations (such as Smith and Minda, 2000; Minda and Smith, 2001) are indeed two decades old. We were unsure whether there was another specific pocket of literature we missed that the reviewers had in mind. If so, we apologize and would be happy to incorporate it.

Also, we felt that there could be more elaboration on the Materials and methods in the main manuscript, specifically as they relate to the theoretical questions at stake. For example, what are the general principles underlying these models and what different hypotheses do they make for this specific stimulus space?

We have added more information on the models and their relationship to the present study to the Introduction:

“We then looked for evidence of prototype and exemplar representations in the brain and in behavioral responses. In behavior, the prototype model assumes that categories are represented by their prototypes and predicts that subjects should be best at categorizing the prototypes themselves, with decreasing accuracy for items with fewer shared features with prototypes. […] We then measured the extent to which prototype- and exemplar-tracking brain regions could be identified, focusing on the VMPFC and anterior hippocampus as predicted prototype-tracking regions, and lateral occipital, prefrontal, and parietal regions as predicted exemplar-tracking regions.”

“To test for evidence of prototype and exemplar representations in behavior across the group, we compared accuracy for items varying in distance from category prototypes and for an accuracy advantage for training items relative to new items matched for distance from category prototypes. […] The model whose predictions better match a given subject’s actual classification responses will have better fit. However, it is also possible that evidence for each of the models will be similar, potentially reflecting a mix of representations.”

8) In addition to engaging more with the literature, we wished for more of a discussion of the mechanistic reasons behind why these ROIs have separate sensitivity to these two models, as well as what the present results show about what underlying computations are occurring in the brain. To quote directly from our consultation session: "I think some attempt at a mechanistic explanation may also be necessary. (Though I understand this can be delicate because they may want to avoid reverse inference.) One thing that stands out to me here is that LO is claimed to be a region showing exemplar representations. LO is believed by many (most?) to be a "lower level" visual area (e.g., in comparison to PPA which may have more abstract information) that represents visual shape information. So even if LO shows an exemplar representation, this could be because the item is more similar in shape to other items that were seen (the exemplars) versus a prototype that was not seen. So, a purely low-level visual account could possibly explain these results, rather than something deeper about mental representations. Thus I am also concerned about whether these findings necessarily mean something deep reconciling this larger theoretical debate, or may reflect some comparison of low level visual features across exemplars." In short, we all agreed that a deeper discussion of the neural computations and mechanisms would improve the contribution of the current paper considerably.

As suggested, we have added more details about the proposed computations performed by individual ROIs and their relevance for prototype and exemplar coding. Introduction:

“Mack and colleagues (2013) found similar behavioral fits for the two models, but much better fit of the exemplar model to brain data. Parts of the lateral occipital, lateral prefrontal and lateral parietal cortices tracked exemplar model predictors. No region tracked prototype predictors. The authors concluded that categorization decisions are based on memory for individual items rather than abstract prototypes. In contrast, Bowman and Zeithamova (2018) found better fit of the prototype model in both brain and behavior. […] However, as neural prototype and exemplar representations were identified across studies that differed in both task details and in the categorization strategies elicited, it has not been possible to say whether differences in the brain regions supporting categorization were due to differential strength of prototype versus exemplar representations or some other aspect of the task.”

Discussion:

“Moreover, our results aligned with those found separately across two studies, replicating the role of the VMPFC and anterior hippocampus in tracking prototype information (Bowman and Zeithamova, 2018) and replicating the role of inferior prefrontal and lateral parietal cortices in tracking exemplar information (Mack et al., 2013). […]The present findings support and further this prior work by showing that regions supporting memory specificity across many memory tasks may also contribute to exemplar-based concept learning.”

Specifically regarding perceptual representations in categorization, there is prior evidence that representations in LO shift as a function of category learning (Palmeri and Gautheir, 2004; Folstein, Palmeri and Gauthier, 2013), which has typically been interpreted as the result of selective attention to category relevant features (Goldstone and Steyvers, 2001; Medin and Schaffer, 1978; Nosofsky, 1986). The study by Mack et al. showed that exemplar representations in LO were not only related to the physical similarity between items, but driven by subjective similarity in individual subjects estimated from their behavioral responses. Thus, these lower level perceptual regions can have strong category effects that are driven by more than just the physical similarity between items. Nonetheless, we note that of our predicted exemplar ROIs, LO showed the weakest evidence of exemplar coding. We have edited the discussion to explicitly point out that one of our hypothesized exemplar regions did not significantly track exemplar predictors. The following is the revised discussion text:

“In addition to IFG and lateral parietal cortex, we predicted that lateral occipital cortex would track exemplar information. […] This aspect of our task may have limited the role of selective attention in the present study and thus the degree to which perceptual regions tracked category information.”

9) Six participants were excluded for high correlated prototype and exemplar regressors – can the authors provide a short of explanation of what pattern of behavioral responses would give rise to this? And what is the average correlation between these regressors in included participants?

There were five participants excluded due to high correlations in one of the learning phases: 3 participants due to high correlation between prototype and exemplar regressor and 2 due to rank-deficient design driven by a lack of trial-by-trial variability in the exemplar predictions. In all five cases, attention weights from their behavioral model fits indicated that the participants ignored most stimulus dimensions in the given phase. As noted in the response to Comment 4A, the models’ predictions are based on estimated perceived distance, which in turn depend on estimated feature attention weights. When too many stimulus features are ignored, too many stimuli are perceived as equivalent and there may not be enough differences between the predictions of the two models (3 subjects) or enough trial-by-trial variability in the model predictions (2 subjects) to permit analysis. We have added a note to the exclusion section providing more explanation regarding the correlation-driven exclusions:

“An additional 5 subjects were excluded for high correlation between fMRI regressors that precluded model-based fMRI analyses of the first or second half of learning phase: 3 subjects had r >.9 for prototype and exemplar regressors and 2 subjects had a rank deficient design matrix driven by a lack of trial-by-trial variability in the exemplar predictor. In all 5 participants, attentional weight parameter estimates from both models indicated that most stimulus dimensions were ignored, which in some cases may lead to a lack of variability in model fits.”

The mean absolute correlation between regressors in included participants was.32. For a detailed response and manuscript edits regarding the regressor correlations for subjects that were retained for analyses, please see Comment 4A.

10) For behavior, the authors investigated performance as it varied with distance from prototypes. It would also be interesting to investigate how behavior varies as distance from studied exemplars.

We agree with the reviewers that, in principle, it is of interest to examine accuracy under the terms of the exemplar model. However, in practice, this can be tricky to accomplish. Because there is only one prototype per category and because the further a stimulus is from one prototype, the closer it becomes to the other prototype, it is easy to display accuracy in terms of the distance from category prototypes. With exemplar representations, that is not the case. There are multiple exemplars per category, so one has to make a choice whether to consider just the closest exemplar or attempt to consider all simultaneously. Taking a linear average of the distance to all exemplars in our category structure would simply be the prototype distance again. The exemplar model uses a nonlinear metric to compute similarity of each item to each exemplar before summing across exemplars, but the exact nature of this nonlinearity is measured for each subject, making it difficult to compile into a group-level analysis. As discussed further in response to Comment 15, getting an aggregate measure of distance/similarity across all training items is an important part of how the exemplar model works. Considering just one closest exemplar provides the best approximation to the prototype analysis, but is necessarily simplified as it does not consider that classifying an item close to two exemplars of one category is easier than classifying an item close to a single exemplar of the category (see also response to Comment 15).

To give the reviewers some sense of what exemplar-based accuracy looks like, we took the approach of calculating the physical distance between a given test item and all the category A and category B training items. We then took the minimum distance to each category (i.e., the physical distance to the closest exemplar from each category), and divided them to compute a “distance ratio.” For example, if the minimum distance to a category A exemplar is 1 (one feature different) and the minimum distance to a category B member is 2 (two features different), then the distance ratio would be 1/2 = 0.5. We focused on the ratio of distances to the two categories because measuring distance to exemplars from only one category does not provide enough information about distance to the other category (unlike the distances to the prototypes, which always sum to 8 and are perfectly inversely related).

The ratio will be 0 for old items themselves as the numerator will be 0 and the denominator would be 4 or 8 given the structure of our training exemplars, either way resulting in zero for the ratio. The ratio will be 1 if the test item is equidistant from the closest exemplar from each category. For all other test items, the ratio will be between zero and one, with numbers closer to zero indicating that a test item is close to an exemplar from one category and far from all exemplars in the opposing category. In practice with our category structure, the possible distance ratio values were 0, 0.2, 0.33, 0.5 and 1. We computed the “proportion correct” metric assuming that category membership is determined by the closer distance. For example, an item with a minimum distance to a category A exemplar of 1 and the minimum distance to a category B member of 2, category A would be considered the correct response as the closest category A item is closer than the closest category B item. We note that for items with a distance ratio of 1 (equidistant), there is no “correct” answer, so we instead represent how often those items were labeled as category A members.

In Author response image 5, panel A shows accuracy as a function of distance ratio during interim tests. Panel B shows accuracy as a function of distance ratio during the final test. What we find is that the gradient based on exemplar distance is less clean than that based on prototype distance. During the interim tests, distance ratios 0-0.5 are clustered together with relatively high accuracy, and do not show an obvious gradient. As would be expected, items that were equidistant between exemplars (distance ratio = 1) are labeled as category A members ~50% of the time after the first test. During the final test, we see that distance ratios 0-0.2 have similar levels of relatively high accuracy, followed by similar levels for ratios 0.3-0.5. Subjects responded category A to about 50% of items with distance ratio = 1, nicely capturing the ambiguity of classifying items equidistant from exemplars from opposing categories.Given the issues noted above with generating any raw-distance-to-exemplars analog to the raw-distance-to-prototype approach, we have elected not to include this analysis in the manuscript.

**Author response image 5. respfig5:** 

11) Did the authors find different weighting on different manipulated features?

We present the attention weights returned by each model averaged across subjects for each phase in Author response table 1:

**Author response table 1. resptable1:** 

Phase	Model	Feature weights (w)	Sensitivity (C)							
		Neck	Tail	Feet	Nose	Ears	Color	Body	Pattern	
Immediate tests 1-2	Exemplar	.12	.15	.16	.18	.10	.11	.10	.08	36.6
	Prototype	.13	.13	.15	.17	.09	.12	.10	.09	24.7
Immediate tests 3-4	Exemplar	.09	.15	.15	.11	.15	.08	.14	.13	56.1
	Prototype	.10	.15	.16	.11	.12	.10	.14	.13	49.3
Final test	Exemplar	.11	.16	.14	.15	.13	.06	.14	.12	57.9
	Prototype	.11	.15	.16	.13	.11	.09	.17	.09	49.7

As there were 8 features and the attention weights sum to 1, a perfectly equal weighing of all features would be.125 weight on each feature. We see that there are some differences in how individual features are weighted, but that the differences are relatively small. It does not seem that there was any feature that was completely ignored by most subjects, nor were there 1-2 features that captured the attention of all subjects. We note that there was also good consistency across the models in how the features were prioritized. We have added the information about these parameters to the data source files in addition to the model fits that were included in the original submission.

12) The manuscript states that for the exemplar model, "test items are classified into the category with the highest summed similarity across category exemplars." One reviewer wondered whether this is a metric that is agreed upon across the field, as they would have anticipated other (possibly non-linear) calculations. For example, could it be quantified as the maximum similarity across exemplars?

Thank you for pointing out that we did not make this clear. Indeed, this is the standard in calculating similarity for the exemplar model to take the sum across similarity to all training exemplars. The probability of responding A is then computed as the relative similarity to each category: the similarity to category A exemplars divided by sum of similarity to category A exemplars and category B exemplars. We adopted this model formalization as used in traditional cognitive modeling literature.

We note, however, that non-linearity is still included in the calculation. Because we (and others) use a nonlinear function to transform physical distance into subjective distance before summing across the items, the most similar items are weighted more heavily than the least similar items. Intuitively, one can recognize a tiger as a mammal because it looks similar to other types of cats, and it does not matter that it does not look like some other mammals, like elephants. Formally, this stems from computing similarity as an exponential decay function of physical distance (so close distances outweigh far distances) rather than a linear function. As a result, two items that are physically 1 feature apart will have a subjective similarity value that is more than twice the similarity value for two items that are 2 features apart. This is canonically how distance/similarity is computed in both models, and is derived from research on how perceived similarity relates to physical similarity (Shepard, 1957).

Although taking the maximum similarity would be a possible function to use for decision making, summing across all exemplars provides an opportunity for multiple highly similar exemplars to be considered in making decisions. For example, imagine a scenario in which there are multiple category A exemplars that are only 1 feature away from a given generalization item vs. a scenario in which there is only one category A exemplar that is 1 feature away (and the other exemplars are 2+ features away). Without summing across the training examples, these two scenarios would generate the same prediction. By summing across them, the model would assign a higher probability of the item being labeled a category A member when there are multiple distance 1 items compared to when there is a single distance 1 item. Because the similarity itself includes a non-linear computation, using the summed similarity function provides the best of both worlds: highly matching exemplars drive the prediction but there is still a differentiation in the resulting probability values informed by other exemplars beyond the closest exemplar.

In response to this comment and also Comment 7, we expanded the model description section in the manuscript to better conceptually explain the similarity computation and the response probability computation. We also made it more clear that these are the canonical ways of computing similarity and response probabilities in these models:

“Exemplar models assume that categories are represented by their individual exemplars, and that test items are classified into the category with the highest summed similarity across category exemplars (Figure 1A). […] This is canonically how an item’s similarity to each category is computed in exemplar models (Nosofsky, 1987; Zaki, Nosofsky, Stanton, and Cohen, 2003).”

13) Is there any way to explore whether the model fits from the neural data can be used to explain variance in participants' behavioral fits? For instance, do neural signals in vmPFC and anterior hippocampus better relate to behavior relative to signals in LO, IFG and lateral parietal for participants with better prototype fits? There may not be sufficient power for this so this was suggested only as a potential exploratory analysis.

In addition to the power issue, we’ve been hesitant to try to relate the neural and behavioral model fits to one another as they are they are not independent. Unlike univariate activation (or another independent neural measure), the brain model fits are computed based on regressors derived from the behavioral model fits and are thus inherently dependent on them. Thus, even if we find a correlation between neural model fits and behavioral model fits, we wouldn’t be able to interpret the neural fits as “explaining” the individuals’ variability in behavior (since the neural fits are computed from behavioral fits). Furthermore, for subjects whose behavior is not well fit by the model, we would be biased *not* to find neural evidence matching the model predictions. Because of this, we’ve been focusing on the main effects (group average) when conducting model-based fMRI analyses in this and our 2018 paper and avoided across-subject brain-behavior correlations.

Out of curiosity, we looked at how the behavioral prototype advantage (the degree of evidence for prototype strategy over exemplar strategy) relates to prototype and exemplar signals in the ROIs using correlations. We did not find any significant correlations, but the strongest relationship found was between behavioral prototype advantage and the VMPFC prototype signal (r = 0.35, p = 0.06 uncorrected), indicating that those who are the strongest “prototypists” tend to have greater prototype signal in VMPFC. This is a nice sanity check, but given the non-independence of the behavioral and neural measures (and the power issues, especially for the number of possible correlations), we did not include this analysis/result in the revised paper. We have however included a note explaining why the neural model fits are inherently linked to behavior and why traditional brain-behavior correlations may not be suitable to use.

“Because each participant’s neural model fit is inherently dependent on their behavioral model fit, we focused on group-average analyses and did not perform any brain-behavior individual differences analyses.”

As a side note, we are currently starting a large individual differences study (now on hold … of course) where we hope to explore the neural predictors of individual differences in strategy, using independent neural markers. So we hope to be able to give a more comprehensive answer to this question in a couple of years…

[Editors' note: further revisions were suggested prior to acceptance, as described below.]

We all agreed that the revised paper was substantially improved. In particular, the removal of the pattern similarity analyses made the paper sharper and more straightforward, as well as eliminated the methodological concerns that were highlighted in our original reviews. The pairwise t-tests of prototype vs. exemplar fits in the ROIs has clarified the results, while still providing good support of the authors' conclusions. The relationship between prototype and exemplar models is much clearer and is thoughtfully discussed, and the added text in the Introduction and Discussion have nicely framed the paper and motivated the analyses. In general, the research reads as being much more impactful and the current work much stronger.Although the paper is much improved, one concern still remains regarding the correlation between exemplar and prototype regressors. To quote reviewer #2 specifically:"I'm convinced that mathematically, the regressors are not highly correlated and the explanation in the authors' response regarding the most distant items is helpful in getting an intuition of how this can be. At the same time, I'm still trying to understand how the correlation between regressors is on average only r=.3 when the behavioral fits themselves are so highly correlated across participants. If there is a way to clarify this, I think that would go a long way in helping readers get on board with the idea that both exemplar and prototype representations can exist in the brain.”

That is correct – high across-subject correlation does not preclude detection of within-subject differences. Intuitively, it is easy to see which students scored better on Midterm 1 vs. Midterm 2, and if one of the Midterms was harder than the other on average, even though the scores on the two midterms would be highly correlated across students. We added an explicit note to clarify that within-subject fit differences are detectable and meaningful even though the fits are highly correlated across subjects.

“Thus, although the model fits tend to be correlated across-subjects, the within-subject advantage for one model over the other is still detectable and meaningful.”

We have also expanded the stimulus set description to incorporate the example of the most distant items that the reviewer found helpful for understanding the model dissociations:

“The stimulus structure enabled dissociable behavioral predictions from the two models. While stimuli near the prototypes also tend to be near old exemplars, the correlation is imperfect. For example, when attention is equally distributed across features, the prototype model would make the same response probability prediction for all distance 3 items. However, some of those distance 3 items were near an old exemplar while others were farther from all old exemplars, creating distinct exemplar model predictions. Because we varied the test stimuli to include all distances from the prototypes, and because within each distance to the prototype there was variability in how far the stimuli are from the old exemplars, the structure was set up to facilitate dissociation between the model predictions.”

What may be less intuitive is that the within-subject correlations of neural model predictions (summed similarity across categories, with an average absolute r=.3 in our data) can be relatively small even when the behavioral predictions (predicted response probabilities) *within that subject* end up similar. An extreme example was observed in Mack et al., 2013, where the behavioral predictions for each of 16 stimuli in their category structure (9 training + 7 transfer) are nearly indistinguishable for the two models and behavior of all but one subject was similarly fit by both models. Yet, neural predictors were largely uncorrelated and neural model fits were reliably more consistent with exemplar model than prototype model in the majority of subjects. As Mack et al. noted, the two models might predict identical behavioral response on any given trial, but the latent representations that support that decision are different, which helps to tease those models apart.

Our category structure was constructed to facilitate distinct behavioral predictions from the two models, so the difference between dissociating behavioral model predictions and dissociating neural model predictions are less staggering in our data. However, to help with intuitive understanding of the neural regressors, we added a conceptual description of the model-based fMRI to the Results section to include explanation of the construction of behavioral vs. neural predictors. We also explicitly noted that representational match to the underlying representations (the neural model predictions) may dissociate the models better than predicted response probabilities (the behavioral model predictions).

“The behavioral model fitting described above maximizes the correspondence between response probabilities generated by the two models and the actual participants’ patterns of responses. […] Furthermore, the neural model fits can help detect evidence of both kids of representations, even if one dominates the behavior.”

Incorporating some of the authors' explanation from the response letter could be useful for this. In particular, it was nice that they explicitly laid out how the advantage for old items over new ones would result in a better fit to an exemplar model but still a decent fit to an exemplar model because accuracy for old/new items at distance 2 fall between accuracy for items with distances of 1 an 3. Their response to Comment 7 seems like it could be informative to include too – although here they talk about the models' differences in terms of confidence, which isn't explained in the manuscript.

Thank you for pointing out that some of the response could be beneficial to include in the paper itself. In the revision, we included the example that the reviewer found helpful (old>new advantage in the context of the overall typicality gradient) as a piece of evidence that both representations may play a role, even when one is more dominant in the behavior:

“To summarize, we observed a reliable typicality gradient where accuracy decreased with the distance from the prototypes and both old and new items at the distance 2 numerically fell between distance 1 and distance 3 items (Figure 3A). However, within distance 2 items, we also observed a reliable advantage for the old items compared to new items, an aspect of the data that would not be predicted by the prototype model.”

Some of the points from Response 7 are also incorporated in the new high-level model-based fMRI overview and reprinted above.

We hope that these revisions have helped the conceptual understanding of the models and how they can help detect both kinds of representations in the brain.